# Potential of Orally Administered Quercetin, Hesperidin, and *p*-Coumaric Acid in Suppressing Intra-/Extracellular Advanced Glycation End-Product-Induced Cytotoxicity in Proximal Tubular Epithelial Cells

**DOI:** 10.3390/ijms26189216

**Published:** 2025-09-21

**Authors:** Takanobu Takata, Junji Moriya, Katsuhito Miyazawa, Sohsuke Yamada, Jia Han, Qian Yang, Xin Guo, Takeshi Nakahashi, Shuichi Mizuta, Shinya Inoue, Togen Masauji, Yoshiharu Motoo

**Affiliations:** 1Division of Molecular and Genetic Biology, Department of Life Science, Medical Research Institute, Kanazawa Medical University, Uchinada 920-0293, Ishikawa, Japan; 2Department of Pharmacy, Kanazawa Medical University Hospital, Uchinada 920-0293, Ishikawa, Japan; masauji@kanazawa-med.ac.jp; 3Department of General Internal Medicine, Kanazawa Medical University, Uchinada 920-0293, Ishikawa, Japan; moriya@kanazawa-med.ac.jp (J.M.); tkn@kanazawa-med.ac.jp (T.N.); smizuta@kanazawa-med.ac.jp (S.M.); 4General Medical Center, Kanazawa Medical University Hospital, Uchinada 920-0293, Ishikawa, Japan; 5Department of Urology, Kanazawa Medical University Hospital, Uchinada 920-0393, Ishikawa, Japan; s-inoue@inoueiin-kusatsu.or.jp; 6Department of Pathology and Laboratory Medicine, Kanazawa Medical University, Uchinada 920-0293, Ishikawa, Japan; sohsuke@kanazawa-med.ac.jp (S.Y.); hanj227@kanazawa-med.ac.jp (J.H.); 7Department of Pathology, Kanazawa Medical University Hospital, Uchinada 920-0293, Ishikawa, Japan; 8Department of Spleen and Stomach Diseases, First Affiliated Hospital of Hebei University of Chinese Medicine, Shijiazhuang 050011, China; yang0311qian@126.com; 9Hebei Key Laboratory of Turbidity Toxin Syndrome, Shijiazhuang 050011, China; tianqi11211216@163.com; 10Research Center, First Affiliated Hospital of Hebei University of Chinese Medicine, Shijiazhuang 050011, China; 11Inoue Iin Clinic, Kusatsu 525-0034, Shiga, Japan; 12Department of Internal Medicine, Fukui Saiseikai Hospital, Wadanaka 918-8503, Fukui, Japan; motoo.yoshiharu9082@fukui.saiseikai.or.jp

**Keywords:** advanced glycation end-product, proximal tubular epithelial cell, traditional Japanese medicine, *Quercus salicina* Blume, *Quercus stenophylla* Makino, quercetin, hesperidin, *p*-coumaric acid, metabolism

## Abstract

Advanced glycation end-products (AGEs) are associated with the dysfunction of proximal tubular epithelial (PTE) cells in lifestyle diseases. Urinary stones induce cytotoxicity in PTE cells, and various medicines have been developed to mitigate or prevent their generation/accumulation. The leaves of *Quercus salicina* Blume/*Q. stenophylla* Makino—used in Japanese folk medicine—contain quercetin, hesperidin, and *p*-hydroxycinnamic (*p*-coumaric) acid, which can suppress the cytotoxicity of intra-/extracellular AGEs. This review investigated the effects of quercetin, hesperidin, and *p*-coumaric acid on PTE cells in terms of their metabolism following oral administration and the associated organs and bacteria. Current evidence indicates that, in PTE cells, non-metabolized quercetin and *p*-coumaric acid may suppress intra-/extracellular AGE-induced cytotoxicity, whereas the metabolites of quercetin and hesperidin may inhibit the generation of AGEs. However, little is known of the effects of *p*-coumaric acid metabolites. Quercetin, hesperidin, and *p*-coumaric acid may collectively suppress the cytotoxicity of intra-/extracellular AGEs in PTE cells. This review on the current paradigm of *Q. salicina*/*Q. stenophylla* extract provides a useful baseline for the design of further preclinical and clinical investigations.

## 1. Introduction

Proximal tubular epithelial (PTE) cells in the kidney regulate the reabsorption of minerals, glucose, amino acids, and raw urine, and regulate the volume of urine [1,2,3]. They are commonly damaged by various types of minerals contained in urinary stones, such as sodium and calcium, inflammation induced by various cytokines and intracellular proteins, and reactive oxygen species (ROS) [1,2,3,4,5]. PTE cell damage and dysfunction have been associated with lifestyle-related diseases (LSRDs) such as diabetes mellitus (DM), cardiovascular disease (CVD), hypertension, steatotic liver disease (SLD), and metabolic dysfunction–associated SLD (MASLD) [6]. Some researchers have revealed mechanisms by which DM induces and promotes the formation of urinary stones [3]. PTE cell death and dysfunction are likely associated with intra-/extracellular advanced glycation end-products (AGEs) [6,7,8,9]. Intracellular AGEs induce inflammation, cell death, and dysfunction in various organs such as the pancreas, liver, brain, and gut [8,9,10,11,12]. In HK-2 cells, a PTE cell line, the generation and accumulation of *N*^ε^-carboxymethyl-lysine (CML)-modified proteins were associated with impaired autophagy and the induction of cytotoxicity, such as inflammation and fibrosis [7]. We hypothesized that various AGEs may modify heat shock protein 90 (HSP90), which is associated with urinary stone adhesion [6]. Various types of AGE receptors are expressed on PTE cells, including the receptor for AGEs (RAGE) [13,14], Toll-like receptor 4 (TLR4) [15,16], and megalin [17,18,19,20]. Extracellular AGEs (e.g., in blood, saline, and urine [8,9,10,11,12]) interact with RAGE and TLR4 to induce inflammation and PTE cell death, whereas megalin stimulates the endocytosis system to promote AGE degradation [13,15,17,19]. Based on various studies of cell survival, PTE cell-related cytotoxicity and dysfunction may be mitigated by inhibiting the accumulation of intracellular AGEs [21,22,23,24,25,26,27,28] and blocking AGE-RAGE/TLR4 signaling [13,14,15,16,17].

Although various medicines have been developed for urinary stone adhesion and inflammation [6], this study focused on the effects of *Quercus salicina* Blume/*Q. stenophylla* Makino leaf extract [29,30,31,32], “Japanese folk medicine,” against modern European and Kampo medicines (traditional Japanese medicine influenced by traditional Chinese medicine) [29,30,31,32,33,34,35]. The aqueous extract of *Q. salicina*/*Q. stenophylla* leaves has been prescribed to remove urinary stones since the early modern period in Japan [29]. In the 20th century, Urocalun was developed based on this folk medicine for traditional and industrial applications [29]. The present review specifically investigates the roles of the components of *Q. salicina*/*Q. stenophylla* leaves—quercetin, hesperidin, and *p*-hydroxycinnamic (*p*-coumaric) acid [29,30,31,32]—in AGE-related cytotoxicity, which have been shown to inhibit (i) the generation of intracellular AGEs [36,37] and (ii) AGE-RAGE/TLR4 signaling [38,39,40,41]. Intracellular AGE generation is suppressed via the “carbonyl trap systems” of quercetin, hesperidin, and *p*-coumaric acid, which require a flavonoid skeleton and a phenolic acid containing a carboxyl group [12,21,22,23]. Accordingly, we hypothesized that quercetin, hesperidin, and *p*-coumaric acid would exert the same effects to protect PTE cells from intra-/extracellular AGE-related cytotoxicity. To test this hypothesis, we predicted and considered various conditions (see Section 5.6 and Section 5.7). We assessed the structural changes in compounds transported into the cells, considering an oral route of administration [29,30,31,32] and that (i) quercetin, hesperidin, and *p*-coumaric acid may not be metabolized or (ii) their metabolites are transported into PTE cells [42,43,44,45,46,47,48]. The metabolites of quercetin and hesperidin, which exhibit a flavonoid skeleton with a carbonyl group, can be transported into PTE cells to inhibit the generation of intracellular AGEs [12]. Similarly, high amounts of *p*-coumaric acid can avoid complete metabolism and be transported into the cells [40] to potentially suppress intracellular AGE generation and AGE-RAGE/TLR4 signaling [11,12]. This review identifies the key advances and gaps in this field, providing a useful reference for further investigation into the clinical applicability and translation of *Q. salicina*/*Q. stenophylla* leaf extract for mitigating PTE cell dysfunction/death and maintaining renal function.

## 2. Urinary Stone/Inflammation in PTE Cells and Roles of *Q. salicina*/*Q. stenophylla* Leaf Extract

### 2.1. Cytotoxicity in PTE Cells

The kidney is the major organ for reabsorbing various essential materials such as minerals, glucose, amino acids, and peptides through the PTE cells [1,2,3,4,5]. PTE cell damage is closely associated with senescence, high urine pressure and volume, excessive minerals, inflammation, ROS, and urinary stone adhesion (Figure 1). Urinary stones can directly damage PTE cells, requiring treatment with approved medicines [3,4,5,6,49,50,51,52]. Urinary stones induce inflammation and the upregulation of proteins associated with apoptosis, such as HSP90 and annexin [4,5,6]. Although the mechanisms by which LSRDs such as DM, CVD, SLD, and MASLD occur and promote urinary stone formation remain unclear, some researchers have reported a relationship between DM and urinary stone production [3]. They noted that sodium-glucose cotransporter-2 (SGLT2) inhibition reduces urinary stone formation. SGLT2 inhibitors increase glycosuria and calcium excretion, which may help explain the mechanisms by which DM promotes urinary stone production. However, whether other LSRDs induce urinary stone formation remains unclear, although they are associated with their generation. In contrast, various molecules can interact with the RAGE and TLR4 expressed on PTE cells to induce inflammation and cell death [53,54,55]. PTE cell damage is thus associated with inflammation from LARDs. Moreover, high intra-/extracellular AGE expression in LSRDs induces PTE cell cytotoxicity and dysfunction (see Section 4) [6].

### 2.2. Medicines Against Urinary Stones

Among the modern European medicines developed from the 18th century, tamsulosin (Astellas Pharma Inc., Tokyo, Japan) [56], magnesium oxide (Maruishi Pharmaceutical Co., Ltd., Osaka, Japan) [57], potassium-/sodium citrate hydrate (Nippon Chemiphar Co., Ltd., Tokyo, Japan) [58], tiopronin (Travere Therapeutics Inc., San Diego, CA, USA) [59], thiazide and thiazide-like diuretics (Merck & Co., Inc., Rahway, NJ, USA) [60], and febuxostat (Teijin Pharma Ltd., Tokyo, Japan) [61,62] have been prescribed for urinary stone adhesion and PTE cell inflammation. These medicines generally comprise one or more low-molecular-weight compounds. Urocalun (Nippon Shinyaku Co., Ltd., Kyoto, Japan)—developed based on the leaf extract of *Q. salicina*/*Q. stenophylla* [29,30,31,63] and Choreito (Tsumura & Co., Tokyo, Japan) [a “Kampo” medicine (see Section 5.1), Chinese name: Zhu-Ling-Tang] [64,65]—has also been prescribed to remove urinary stones from the proximal tubule. Although Urocalun is currently prescribed in clinical, hospital, and pharmaceutical practice in Japan, *Q. salicina*/*Q. stenophylla* leaf extract is applied largely as a folk medicine.

## 3. Intra-/Extracellular AGEs

### 3.1. Origins of AGEs

Protein glycation occurs commonly in vivo and in vitro [66,67,68]. Although the proteins in, for example, glycated hemoglobin (HbA1c) can be recovered, the saccharides in AGEs cannot be recovered along with their metabolites/non-enzymatic products and protein molecules [6,7,8,9,66,67,68]. We investigated the origin of AGE-related glucose, fructose, glyceraldehyde, glycolaldehyde, glyoxal, methylglyoxal, and 3-deoxyglucosone (Figure 2) [6,9,11,12]. Glyceraldehyde, glycolaldehyde, glyoxal, and methylglyoxal are produced from saccharides (e.g., glucose, fructose) and lipids [6,69,70,71,72,73]. Because AGEs and advanced lipoxidation end-products (ALEs) are generated from the same molecules, some AGEs and ALEs have the same characteristics [6,12]. Therefore, the excess intake of saccharides and lipids can induce LSRDs via the generation of AGEs and ALEs [6,12]. This suggests that AGEs and ALEs require more in-depth classification.

### 3.2. Structure and Category of AGEs

#### 3.2.1. Structure of Free AGEs

AGEs are generated from saccharides and their metabolites/non-enzymatic products and proteins, and they react and combine with proteins through non-enzymatic reactions [6,9,11,12]. Accordingly, AGEs are considered modified proteins. A “free type of AGE” can be generated through amino acid reactions, including CML, *N*^ε^-carboxyethyl-lysine (CEL), methylglyoxal-hydro-imidazolone (MG-H1), and methylglyoxal-lysine dimer (MOLD) [6,9,10,11,12], several of which have been detected in humans/animals (e.g., organs, cells, blood, and urine), as well as in foods and beverages (Figure 3) [6,9,10,11,12]. AGEs have also been artificially synthesized (e.g., trihydroxy-triosidine, lysine-hydroxy-triosidine, arginine-hydroxy-triosidine, and a pyrrolopyridinium lysine dimer derived from glyceraldehyde) [74,75]. In 2023, Takeuchi et al. [76] reported two hypothetical structures for glyceraldehyde-derived AGEs (GA-AGEs), which they named “Toxic AGEs (TAGE)” in 2004; however, the structures have not been proved (see Section 3.2.2).

#### 3.2.2. Classic and Novel AGEs

Studies on AGE catalyzation since the mid-20th century have identified glucose-derived (Glc-AGEs, AGE-1), GA-derived (AGE-2), glycolaldehyde-derived (Glycol-AGEs, AGE-3), methylglyoxal-derived (MGO-AGEs, AGE-4), glyoxal-derived (GO-AGEs, AGE-5), and 3-deoxyglucosone-derived AGEs (3DG-AGEs, AGE-6) [9,11,12]. However, novel types of AGEs with different compositions may require novel categories. Lactaldehyde-derived and melibiose-derived AGEs (MAGE) were recently synthesized [77,78]. Litwinowicz et al. [78] identified MAGEs in the blood of patients with alcoholic liver disease, which was categorized as AGE-10. Because the MGO-AGEs MG-H1 and argpyrimidine can be generated from glyceraldehyde and methylglyoxal, they should be categorized as GA- and MGO-AGEs [11,12]. Sugawa et al. [79] reported that CML can be generated from ribose via the production of glyoxal. Moreover, although glucoselysine is produced from glucose, it can also be produced from fructose [80,81]. Various researchers have proposed other categorizations based on cytotoxicity. For example, Takeuchi et al. [76] defined some GA-AGEs as “TAGE” in 2004 based on their recognition by in-house-prepared polyclonal anti-AGE antibodies, though the structures of the TAGEs remain unclear. Simkova et al. [82] classified GA- and Glycol-AGEs containing pyridinium moieties as TAGEs. Shen et al. [83] categorized GA-, MGO-, GO-, and 3DG-AGEs as TAGEs. In contrast, Lee et al. [6,84] categorized MOLD (one of the MGO-AGEs), which contains two amino acids and has been confirmed in humans, as a TAGE that may be associated with LSRDs (Figure 3b). Various researchers have defined TAGEs based on their cytotoxicity and associations with LSRDs [6,76,82,83,84].

#### 3.2.3. Crude AGE Pattern

Various AGE structures may be generated from one source compound [85]. Although this phenomenon has been demonstrated using electrospray ionization (ESI)-mass spectrometry (MS) (ESI-MS) and matrix-assisted laser desorption/ionization-MS (MALDI-MS) in vitro (in cultured cells) and in tube experiments, no report has named or categorized this phenomenon. However, understanding this phenomenon may be important for future animal models and clinical investigations. Therefore, we recently termed it a crude AGE pattern [6,11,12]. In previous reports, Senaviralthna et al. [85] showed that MG-H1-, argpyrimidine-, and glyceraldehyde-derived pyridinium (GLAP)-modified proteins were generated in human pancreatic ductal epithelial PANC-1 cells treated with glyceraldehyde using ESI-MS (Figure 4). MG-H1- and argpyrimidine-modified HSP90 have been artificially generated through incubation of methylglyoxal and HSP90 at 37 °C [6].

#### 3.2.4. Diverse AGE Patterns

Amino acid residues such as lysine and arginine are targets of non-enzymatic glycation, and proteins can be modified by various free types of AGEs in vitro and in vivo [86,87]. This has been demonstrated by researchers who analyzed AGEs using ESI-MS and MALDI-MS. These findings suggest that researchers should focus on the diverse types of AGEs that modify proteins, rather than on a single type of AGE modification, when analyzing AGE-induced cytotoxicity such as protein dysfunction. In contrast, a single type of AGE can modify various proteins, and these phenomena have been generally recognized [11,12]. However, these phenomena have not been named or categorized. Therefore, we proposed type 1 and 2 diverse AGE patterns (Figure 5) [11,12,88]. In the type 1 diverse AGE pattern, a certain AGE structure can modify a certain protein (1A: different types of AGE structures modified by the same amino acid residue; 1B: same type of AGE structures modified by different amino acid residues; 1C: different types of AGE structures modified by different amino acid residues; Figure 5a). These patterns can be validated using MS such as ESI-MS and MALDI-MS [11,12,87,88]. In the type 2 diverse AGE pattern, one type of AGE structure can modify various proteins (Figure 5b), which can be validated through Western blot analysis using anti-AGE antibodies.

#### 3.2.5. Multiple AGE Patterns

Based on the type 1 and 2 diverse AGE patterns, we proposed type 1 and 2 multiple AGE patterns, as these phenomena have not yet been named or categorized in the field of AGE research (Figure 6) [11,12,87,88]. These patterns can be validated using MS such as ESI-MS and MALDI-MS [6,11,12,87,88]. In the type 1 multiple AGE pattern, certain AGE structures can modify one protein molecule but not a specific type of protein (Figure 6a). In the type 2 multiple AGE pattern, the AGE structure is modified by more than two proteins through intermolecular covalent bonds (Figure 6b).

### 3.3. Intracellular AGEs

Various organs such as the heart, liver, pancreas, and proximal tubule express intracellular AGEs, with crude, diverse, and multiple AGE patterns [11,12]. Intracellular AGEs can be generated and accumulated in normal human tissue, especially in patients with LSRDs. 2-Ammnonio-6-[4-(hydroxymethyl)-3-oxidopyridinium-1-yl]-hexanoate-lysine (4-hydroxymethyl-OP-lysine) modified with ryanodine receptor 2 (RyR2) and MG-H1 and G-H1 modified with the F-actin tropomyosin filament can induce beating dysfunction in cardiomyocytes [12]. The accumulation of CEL in skeletal muscles can promote the formation of adipose tissue [9,12]. Glucoselysine in the lens may promote cataract formation in model rats [80,81].

### 3.4. Extracellular AGEs

#### 3.4.1. AGEs in the Blood, Saliva, and Urine from Various Organs

Once intracellular AGEs are generated in various organs, they can disperse and be secreted into the blood, saliva, and urine [9,11,12]. Some cytotoxic AGEs may leak into the body fluid, where they induce cytotoxicity and cell dysfunction. These AGEs interact with RAGE and TLR4 expressed on the cell membranes of various organs to induce cytotoxicity and inflammation. AGE-RAGE/TLR4 signaling is closely related to LSRD pathogenesis. Megalin expressed in the proximal tubule activates the endocytosis system to degrade AGEs. Since extracellular AGEs reflect the dynamics of intracellular AGEs, they are useful biomarkers of organ dysfunction and disease development [9,11,12]. Litwinowicz et al. [78] reported a novel MAGE biomarker for alcoholic liver disease. Katsuta et al. [89] found that blood CML and MG-H1 levels may be associated with kidney function after transplantation. The AGE structures can be determined through enzyme-linked immuno-sorbent assay (ELISA) and MS, such as ESI-/MALDI-MS.

#### 3.4.2. AGEs in Extracellular Matrix

AGEs such as pentosidine can exist in the extracellular matrix [11,12,88]. Pentosidine-modified collagen tissue exhibits various dysfunctions compared with normal collagen tissue. We believe that intracellular collagens may be modified by AGEs and subsequently released.

#### 3.4.3. Dietary AGEs

AGEs such as CML, CEL, and MG-H1 are produced in various foods and beverages, and can be consumed by humans [9,11,12,88]. These AGEs interact with RAGE and TLR4 in various organs as well as AGEs in the body fluid. The AGEs in the blood and urine may closely reflect diet and lifestyle.

### 3.5. Identification and Quantification of AGEs

The identification and quantification of AGEs are generally performed through fluorescence microscopy [10,11,12], Western blot [10,11,12,25], slot blotting [10,11,12,25,88,90], ELISA [10,11,12,91], MS such as gas chromatography-MS (GC-MS), ESI-MS, MALDI-MS [10,11,12,90], and nuclear magnetic resonance (NMR) [11,12,77]. Fluorescence microscopy and high-performance liquid chromatography (HPLC) are suitable since AGEs emit fluorescence [10,11,12]. Western blotting, slot blotting, and ELISA use anti-AGE antibodies to detect AGEs, though they do not provide structural information [10,11,12,25,90]. Standard AGE-modified proteins are prepared in test tubes, whereas normal recombinant proteins can be prepared in cells [10,88,92]. In the quantification, we recommend that slot blotting be performed based on Takata’s method, using Takata’s lysis buffer (modified or not) to stably probe AGE-modified proteins onto polyvinylidene fluoride (PVDF) membranes [10,25,88,90]. Therefore, this method was selected in thirteen studies from 2017 to 2025 [10,25,90]. For MS, quantification must be performed for AGEs with defined structures and with appropriate standards [10,11,12,89]. For quantification, GC-, ESI-, and MALDI-MS systems can be connected with HPLC platforms. NMR can identify AGEs but not quantify them [10,12,77,78]. Western blotting using only anti-AGE antibodies is inappropriate for detecting type 1 diverse and type 1 and 2 multiple AGE patterns; ESI-/MALDI-MS must instead be performed (Figure 5a and Figure 6) [6,11,12,88].

## 4. Cytotoxicity of Intra-/Extracellular AGEs in PTE Cells

### 4.1. Cytotoxicity of Intracellular AGEs in PTE Cells

Takahashi et al. [7] suggested that the generation/accumulation of intracellular AGEs (likely various Glc-AGEs) in a human PTE cell line (HK-2 cells) promoted the production of lysosomes to degrade AGE-modified proteins. They revealed that CML-modified proteins increased in HK-2 cells in which Atg5 was knocked down and incubated in high-glucose medium [7]. They insisted that intracellular AGE-modified proteins increase in the proximal tubular cells, where the dysfunction of autophagy is induced, and promote inflammation and fibrosis of cells. Although they analyzed only CML-modified proteins in HK-2 cells with Western blot, other types of AGE-modified proteins might be generated in these cells in their study because glucose is the origin for various types of AGEs (e.g., CEL) [9,11,12]. In contrast, the role of HSP90 as a receptor for urinary stones in HK-2 cells may be modified by various types of AGEs based on diverse and multiple AGE patterns [6]. Since MG-H1 and argpyrimidine-modification may cause dysfunction in human recombinant HSP90, we hypothesized that HSP90 expressed in tubular epithelial cells is modified by these AGEs under high-glucose and/or fructose conditions [6]. Current evidence suggests that various types of AGEs may be generated in PTE cells depending on the crude, diverse, and multiple AGE patterns (Figure 4, Figure 5 and Figure 6).

### 4.2. Cytotoxicity of Extracellular AGEs in PTE Cells

RAGE [13,14], TLR4 [15,16], and megalin [17,18,93] are expressed on the surface of PTE cells. Although megalin participates in the endocytosis of AGEs from the blood and urine [17,18], the AGE-RAGE/TLR4 axis induces cytotoxicity and inflammation in various organs such as the liver, heart, lung, and gut [9,13,14,15,16]. Chen et al. [14] reported that activation of AGE-RAGE signaling in the kidney upregulated the expression of tumor necrosis factor alpha (TNF-α), interleukin (IL)-1 beta (IL-1β), IL-6, and IL-18 in renal tissues, and increased the blood levels of these cytokines. AGE-RAGE signaling strongly activates the TLR4/myeloid differentiation factor 88 (MyD88)/nuclear factor-κβ (NF-κβ) pathway to exacerbate inflammation [14]. AGE-TLR4 interactions also induce inflammation [9]. Although urinary stones adhere to receptor proteins such as HSP90, annexins [4,5], and CD44 in PTE cells to induce inflammation, the roles of the AGE-RAGE/TLR4 axis in inflammation require further investigation. Finally, the soluble receptor for AGEs (sRAGE)—a cleaved-type RAGE released/secreted from cell membranes in the blood and urine—can also react with AGEs and potentially inhibit the cytotoxic effect associated with AGE-RAGE interactions [94].

## 5. Potential of Quercetin, Hesperidin, and *p*-Coumaric Acid for Suppressing Intra-/Extracellular AGE-Induced Cytotoxicity in PTE Cells

### 5.1. Q. salicina/Q. stenophylla Leaf Extract

Modern European and traditional medicines are commonly used in Japan. Traditional Japanese medical science often adopts Kampo [32,33,34,95,96] and folk medicines [29,30,31]. Traditional Chinese medical science involves a variety of therapeutic approaches, such as health management (diet and exercise) [97], crude drugs (natural medicines) [33,34,95,96,98,99], moxibustion/cupping treatment [32,33,34,95,96,99], and acupuncture [100,101,102], and was introduced in ancient Japan from the Chinese mainland and Korean peninsula and modified to establish Kampo (“Kam” or “Kan” in Japanese means “Han” in Chinese, referring to the ancient Chinese empire; “po” refers to the method of diagnosis or treatment; Figure 7). Kampo medicines contain multiple crude drugs, whereas original Japanese medicine is prepared using one crude drug (e.g., specific plant extract). Although much of traditional Japanese medical science is considered Kampo, Japanese folk medical science traditionally incorporates crude plant extracts [29,30,31]. *Q. salicina*/*Q. stenophylla* leaf extract has been used in folk medicine to remove urinary stones from the proximal tubules since the early modern period in Japan. It is unclear whether this extract was used as a regular authentic medicine in other areas such as China, Korea, and Vietnam, as this information did not reach Japan during the ancient to early modern periods. Therefore, we believe that *Q. salicina/Q. stenophylla* leaf extract is one of the Japanese folk medicines. The most important contemporary derivative, Urocalun, was only developed from the aqueous extract of *Q. salicina*/*Q. stenophylla* in the 20th century [29,30,31].

### 5.2. Limited Data on the Effects of Q. salicina/Q. stenophylla Leaf Extract Components

We focused on the possibility that medicines selected to prevent and treat urinary stones may also suppress intra-/extracellular AGE-induced cytotoxicity in PTE cells. However, no study has assessed the inhibitory effects of modern European medicines such as tamsulosin [56], magnesium oxide [57], potassium-/sodium citrate hydrate [58], tiopronin [59], thiazide and thiazide-like diuretics [60], and febuxostat [61,62] and traditional medicines such as Urocalun [29] and Choreito [64,65] on the generation of intracellular AGEs or AGE-RAGE/TLR4 signaling. Demonstrating that these urinary stone medicines exert anti-AGE effects in PTE cells would provide important information for the treatment and prevention of PTE cytotoxicity. Although the chemical structures of modern European medicines have been identified, their potential anti-AGE effects remain unclear [56,57,58,59,60,61,62]. Urocalun and Choreito contain multiple components, some of which have been identified, and they are known to include anti-AGE compounds. At this stage, we focused on the components of *Q. salicina*/*Q. stenophylla* leaf extract, which are major constituents of Urocalun. They are categorized as Japanese folk medicines rather than Kampo medicines, as research on folk medicines has not been as actively promoted compared with Kampo medicines. Accordingly, little is known of the effects of *Q. salicina*/*Q. stenophylla* leaf extract on AGE expression and signaling, despite its widespread adoption along with Urocalun. Because of the lack of studies, we characterized the effects of each compound based on the available mechanistic research. Therefore, direct treatments using the extract are unsuitable for describing the in vitro effects of the various components, whereas treatments with the respective components or their metabolites may effectively elucidate their effects. This approach is especially important given the metabolic processes that natural compounds and their metabolites are exposed to during transport to PTE cells following oral administration.

### 5.3. Quercetin, Hesperidin, and p-Coumaric Acid

Quercetin, hesperidin, and *p*-coumaric acid have been detected in various extracts of *Q. salicina*/*Q. stenophylla* (Figure 8) [6,29,30,31,32]. Quercetin [103,104], hesperidin [104,105,106,107], and *p*-coumaric acid [108,109] are produced by and accumulate in various plants. Although quercetin and hesperidin are flavonoids, the former has an aglycon structure and the latter has a glycoside structure [12,110,111]. Hesperetin-7-*O*-rutinoside, because hesperetin (aglycon-structured hesperidin) is modified with two hexoses. Although various flavonoids (e.g., rutin, naringin, quercetin, hesperidin, and myricetin) and phenolic acids (e.g., gallic acid, pyrogallol, caffeic acid, *p*-coumaric acid, and *m*-coumaric acid) have been isolated [32], we selected quercetin and hesperidin as typical examples of aglycon- and glycoside-type flavonoids, and *p*-coumaric acid as an example of phenolic acid. Although Aung et al. [32] characterized the leaf contents (μg/g) of various natural compounds such as flavonoids and phenolic acids, this study focused only on structure.

### 5.4. Suppression of Intracellular AGE Generation by Quercetin, Hesperidin, and p-Coumaric Acid in Cells Other than PTE Cells

Various studies have investigated the effects of low-molecular-weight plant compounds in suppressing intracellular AGE accumulation [112]. Quercetin, hesperidin, and *p*-coumaric acid have each been reported to inhibit the generation of intracellular AGEs through the molecular suppression of AGE generation [6,12,21,22,23,24,25,26]. The process is primarily facilitated through the “carbonyl trap system” (not the activation of glyoxalase [6,12,36,37]), which targets the source compounds such as glyceraldehyde, glycolaldehyde, methylglyoxal, glyoxal, and 3-deoxyglucosone [6,12]. The carbonyl trap system targets the ketone and aldehyde groups of these source compounds and functions as a non-enzymatic reaction (a simple organic chemical reaction that does not require enzymes). Many experiments have demonstrated that the carbonyl trap system can inhibit the generation of AGEs in vitro and in tube [6,12]. Quercetin, hesperidin, and *p*-coumaric acid may suppress the generation of various free AGEs, as well as diverse and multiple AGE patterns. The effects of quercetin and hesperidin are specifically determined by the flavonoid skeleton [110,111,113,114].

### 5.5. Inhibition of Extracellular AGE-RAGE/TLR4 Cell Signaling by Quercetin, Hesperidin, and p-Coumaric Acid in Cells Other than PTE Cells

To suppress cytotoxicity-related inflammation, various studies have investigated the effects of low-molecular-weight plant compounds on AGE-RAGE/TLR4 cell signaling [9,12]. Quercetin has been found to suppress the AGE-RAGE axis [38,115]. Quercetin-induced suppression of TLR4/NF-κβ signaling has been shown to inhibit inflammation and cell death in vitro and in vivo [116,117,118,119,120]. Therefore, we believe quercetin can suppress RAGE/TLR4-related cytotoxicity. However, because quercetin can exert cytotoxicity through RAGE, the inhibitor for the AGE-RAGE/TLR4 axis should be considered carefully [121,122]. Hesperidin can suppress RAGE/NF-κβ and TLR4/NF-κβ signaling in vitro and in vivo [39,123,124,125]. Various studies have indicated that *p*-coumaric acid can suppress the AGE-RAGE axis in vitro and in vivo [40,126]. Because *p*-coumaric acid can also suppress inflammation via TLR4/NF-κβ signaling in vivo [41,127], it may inhibit AGE-RAGE/TLR4-induced inflammation. The suppressive effects of quercetin, hesperidin, and *p*-coumaric acid on cytotoxicity via RAGE/TLR4 signaling have been demonstrated in non-clinical investigations.

### 5.6. The Issue of Whether Quercetin, Hesperidin, and p-Coumaric Acid Can Affect PTE Cells

There are no reports that orally administered quercetin, hesperidin, or *p*-coumaric acid suppress cytotoxicity of PTE cells induced by intra-/extracellular AGEs in vitro, in vivo (animal models), or in patients. Although these three compounds may inhibit the generation of intracellular AGEs and suppress extracellular AGE-RAGE/TLR4 signaling in PTE cells in vitro, whether these functions are exhibited in vivo or in clinical investigations remains unclear. However, (i) the carbonyl trap system can act regardless of cell type because it is a simple organic chemical reaction [6,12], and (ii) RAGE and TLR4 are expressed on PTE cells, with downstream NF-κβ signaling similar to other cell types [13,14,15,16]. Therefore, these compounds may protect PTE cells against intra-/extracellular AGE-induced cytotoxicity in vitro and in vivo. To hypothesize that orally administered quercetin, hesperidin, and *p*-coumaric acid protect PTE cells from AGE-induced cytotoxicity, we predicted and considered various conditions, as shown below. If quercetin, hesperidin, and *p*-coumaric acid cannot be transported into PTE cells, they will be unable to exert their anti-cytotoxic effects. This is an important issue that must be clarified to determine whether orally administered quercetin, hesperidin, and *p*-coumaric acid can exert anti-AGE effects in PTE cells in vivo and in patients. Based on this theme, we also need to consider the possibility that these compounds, or their metabolites, are transported into PTE cells and exert anti-AGE effects, since orally administered compounds are absorbed and metabolized [9].

### 5.7. Transportation Potential of Non-Metabolites and/or Metabolites of Quercetin, Hesperidin, and p-Coumaric Acid into PTE Cells

#### 5.7.1. Absorption and Metabolization of Quercetin

A microbalance of quercetin has been detected in human blood [128]. This suggests that quercetin (i) can resist various metabolic systems and (ii) maintain the aglycon structure following metabolism by various enzymes and bacteria in different organs. Various quercetin metabolites have been recorded in human blood [129,130,131]. Although whole-metabolite profiles remain elusive, quercetin-3-*O*-glucoside (Q3G), quercetin 3-*O*-glucuronide (Q3GA), 3′-*O*-methylquercetin-*O*-glucuronide (3′-*O*-methyl-Q3GA), quercetin-4′-*O*-glucoside (Q4′G), quercetin-4′-O-glucuronide (Q4′GA), 3′-*O*-methylquercetin-4-*O*-glucuronide (3′-*O*-methyl-Q4′GA), quercetin-3′-glucuronide (Q3′GA), and quercetin-3′-O-sulfate (Q3S) have been reported in human blood (Figure 9) [129,130,131]. Quercetin can be metabolized into structures such as glycosides, glucuronides, and sulfates, though mainly non-metabolized quercetin is transported into PTE cells. Q3G and Q4′G can be transported to the small intestine via SGLT1 (Figure 10) [132,133,134]. Q3G and Q4′G are hydrolyzed via lactase phlorizin hydrolase (LPH) expressed on the membrane of small intestinal epithelial cells to produce quercetin, which can then be absorbed into PTE cells (Figure 10) [134,135,136]. Intracellular Q3G and Q4′G are hydrolyzed by β-glucosidase to produce quercetin, which is metabolized to glucuronide via phase II enzyme systems (Figure 10) [136]. Although little is known of quercetin-glucuronides, Q3GA, 3′-*O*-methyl-Q3GA, Q4′GA, 3′-*O*-methyl-Q4′GA, and Q3′GA may be produced in this step. Quercetin-glucuronides are dispersed into the lymph fluid and blood from the large intestine epithelial and hepatic parenchymal cells [136]. Quercetin-glucosides are transported from the small intestine into the large intestine, where bacteria hydrolyze them to produce quercetin [137,138]. The transport potential of quercetin has been confirmed in human colon Caco-2 cells (Figure 11) [139,140,141]. Murota et al. [142] reported that Q3G and Q4′G were likely absorbed into Caco-2 cells via SGLT1 as the transporter (Figure 11). In the large intestine, glucuronide-quercetins may be produced via β-galactosidase and phase II enzymes, and subsequently leak into lymph fluid and blood. We assessed the hepatic metabolism of quercetin [143,144,145,146,147]. Quercetin has been shown to protect against fibrosis and lipogenesis in hepatic tissue [143,144]. Kaci et al. [145] predicted that G3S may be produced as a metabolite. Hou et al. [146] suggested that quercetin metabolites may protect hepatic cells. Michala et al. [147] detected 3′-*O*-methylquercetin-7-*O*-glucuronide (3′-O-methyl-Q7GA) and 4′-*O*-methylquercetin-7-*O*-glucuronide in human hepatic parenchymal HepG2 cells treated with quercetin (4′-O-methyl-Q7GA) (Figure 12).

#### 5.7.2. Absorption and Metabolization of Hesperidin

Hesperidin has never been recorded in the blood of humans, rats, or mice, suggesting that it cannot escape metabolism. Although hesperidin cannot be hydrolyzed by α-glucosidase in the small/large intestine, it can be hydrolyzed to hesperetin by bacterial β-glucosidase and absorbed into the small/large intestine (Figure 13) [148]. Hesperetin in the small/large intestine can be metabolized via the phase II enzyme system to hesperetin-glucuronide. Hesperidin can be transported into the small [149,150] and large intestine (Figure 13) [151,152]. Brand et al. [152] reported that UDP-glucuronosyltransferases (UGTs) and sulfotransferases (SULTs) in the human small/large intestines and liver can metabolize hesperidin. In the cytosolic fractions of these tissues, hesperetin-3′-*O*-glucuronide, hesperetin-7′-*O*-glucuronide, hesperetin-3′-*O*-sulfate, and hesperetin-7′-*O*-sulfate were identified following incubation with hesperidin (Figure 14) [151,152]. Hesperidin has been recorded in the liver of animal models, and its metabolites in a liver with fatty-liver characteristics suppressed inflammation and fibrosis [153,154]. In human hepatic parenchymal Hepa-RG and stellate LX-2 cells, hesperidin exerted antioxidant effects by increasing the protein levels of copper/zinc superoxide dismutase (SOD1), manganese superoxide dismutase (SOD2), and catalase, and by suppressing lipid oxidation [155]. These effects may have been associated with the anti-lipidation activities of its metabolites.

#### 5.7.3. Absorption and Metabolization of *p*-Coumaric Acid

*P*-coumaric acid has been detected in human blood and can be absorbed into the small/large intestine because of its resistance to enzymatic and bacterial metabolism [156,157]. In plants, *p*-coumaric acid can be metabolized by some enzymes such as 4-coumaric CoA-ligase (4CL), hydroxycinnamoyl transferase (HCT), *p*-coumaroyl-shikimic acid-3-hydroxytransferase (C3′H), and caffeoyl shikimic esterase (CSE) to produce caffeic acid (in this pathway, *p*-coumaroyl-CoA, *p*-coumaroyl-shikimic acid, and caffeoyl-shikimic acid are produced as internal metabolites) (Figure 15) [158,159]. However, the metabolic dynamics of *p*-coumaric acid in the human body remain unclear. Because high amounts of *p*-coumaric acid are received with oral administration and detected in the blood, it is likely that small amounts may be metabolized owing to slight structural variation.

#### 5.7.4. Transportation of Non-Metabolites or Metabolites of Quercetin, Hesperidin, and *p*-Coumaric Acid into PTE Cells

This study focused on the capacity of quercetin, hesperidin, *p*-coumaric acid, and their metabolites for transport into PTE cells. Based on their predicted structures, current evidence indicates that the metabolites of quercetin and hesperidin can be transported into PTE cells. Because quercetin and *p*-coumaric acid have been detected in human blood [129,156,157], the non-metabolized forms may be transported into PTE cells [although some molecules may be metabolized in the small or large intestine, the metabolites may eventually revert to their original structure (Figure 10 and Figure 11)]. In contrast, the evidence suggests that hesperidin cannot be transported into PTE cells because it has never been detected in human blood. Quercetin, quercetin-glucoside, and hesperidin are metabolized in the small/large intestine and liver, and their metabolites can also be transported into PTE cells (Figure 10, Figure 11, Figure 12, Figure 13 and Figure 14). Because *p*-coumaric acid shows resistance to enzymatic and bacterial metabolism [156,157], we predict that, along with its metabolites, some *p*-coumaric acid may be transported into PTE cells. In contrast, we cannot introduce the metabolites of *p*-coumaric acid.

#### 5.7.5. Potential for Suppressing Intra-/Extracellular AGE-Induced Cytotoxicity by Non-Metabolites/Metabolites of Quercetin, Hesperidin, *p*-Coumaric Acid in PTE Cells

Based on Section 5.4 and Section 5.5, we hypothesize that orally administered quercetin and *p*-coumaric acid can suppress intra-/extracellular AGE-induced cytotoxicity in PTE cells, provided they are effectively transported into the target cells while maintaining their non-metabolic structures, although the metabolic processing of quercetin and *p*-coumaric acid requires further characterization [21,22,23,24,25,26,27,28,156,157]. Because the inhibition of intracellular AGEs by the carbonyl trap system of quercetin and *p*-coumaric acid can occur regardless of cell type, we believe they may inhibit intracellular AGE formation in PTE cells. Most hesperidin molecules may be metabolized before being transported; thus, it is unclear whether they participate in the suppression of intra-/extracellular AGE cytotoxicity in PTE cells. In contrast, the metabolites of quercetin and hesperidin may exert inhibitory effects owing to their robust flavonoid skeleton. Indeed, many flavonoid-based natural products have been shown to suppress intracellular AGE generation (Figure 8, Figure 9, Figure 10, Figure 11, Figure 12, Figure 13 and Figure 14) [12]. However, whether the metabolites of quercetin and hesperidin suppress cytotoxicity via the AGE-RAGE/TLR4 axis is unclear. In particular, further characterization of the metabolites of *p*-coumaric acid is needed to determine their anti-AGE effects in humans. If *p*-coumaric acid is able to be metabolized to caffeic acid in the human body, and it can be transported into PTE cells, it may inhibit the generation/accumulation of intracellular AGEs (Figure 15) [160,161].

## 6. Limitations

The complete composition of *Q. salicina*/*Q. stenophylla* leaf extract remains unclear, and the contents of quercetin, hesperidin, and *p*-coumaric acid may fluctuate depending on the collection site and year. This is a limitation compared with modern European medicines. Although we considered the metabolism of quercetin, hesperidin, and *p*-coumaric acid in the small and large intestine and liver, we did not provide a detailed section on pharmacokinetic data, nor did we include information comparing plasma and urine levels after oral dosing in humans or animals, metabolite profiles, renal-specific transport, the influence of gut bacteria, or bioactivity. The study neglected some factors, such as physiological reactions in the mouth and gut [162], where quercetin and hesperidin may be glycosylated and hydrolyzed, which should be addressed in a future review. Additionally, whether these compounds are present at pharmacologically relevant levels for the inhibition of intra-/extracellular AGE-related cytotoxicity requires experimental verification.

## 7. Conclusions

This study indicated that orally administered quercetin, hesperidin, and *p*-coumaric acid may suppress intra-extracellular AGE-mediated cytotoxicity in PTE cells. The non-metabolites of quercetin and *p*-coumaric acid and the metabolites of quercetin and hesperidin inhibit intracellular AGE generation via the carbonyl trap system. Although various types of AGEs are generated in PTE cells, the carbonyl trap system targets important source components. The non-metabolites of quercetin and *p*-coumaric acid modulate AGEs-RAGE/TLR4 signaling. However, the functions of the metabolites of quercetin, hesperidin, and *p*-coumaric acid in AGEs-RAGE/TLR4 signaling remain unclear. This study provides a useful basis on which to design future preclinical and clinical research on the performance and mechanisms of *Q. salicina*/*Q. stenophylla* leaf extract in suppressing intra-/extracellular AGE-induced cytotoxicity and treating related urinary disorders.

## 8. Future Directions

To advance the field, future in vitro and in vivo studies should consider how and to what extent quercetin, hesperidin, *p*-coumaric acid, and the other components of *Q. salicina*/*Q. stenophylla*, along with their metabolites, inhibit the generation/accumulation of intracellular AGEs and suppress the signaling of extracellular AGEs-RAGE/TLR4 in PTE cells. Subsequently, the effects of orally administered *Q. salicina*/*Q. stenophylla* leaf extract should be determined in animal models presenting with intra-/extracellular AGE-induced cytotoxicity in PTE cells.

## Figures and Tables

**Figure 1 ijms-26-09216-f001:**
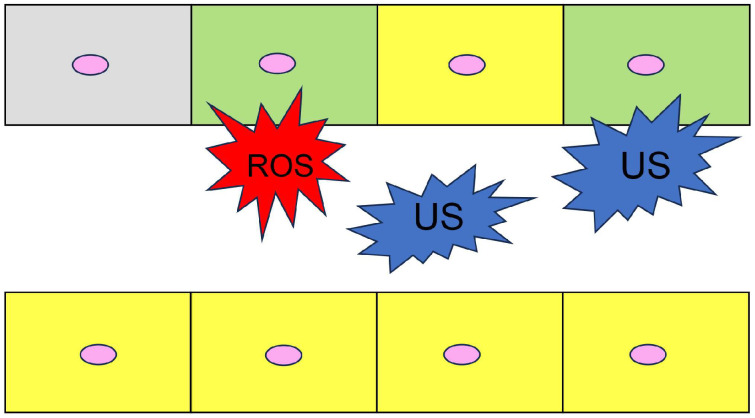
Proximal tubular epithelial (PTE) cells and mechanisms of dysfunction. Upper and lower cells form the proximal tubular walls, and the middle space is the urinary tract. Closed squares indicate PTE cells. Closed pink circles indicate cell nuclei. Closed red polygon indicates reactive oxygen species (ROS). Closed blue polygons indicate urinary stones. Yellow cells indicate the normal cells. Gray cell indicates a senescent cell. Green cells indicate cells with inflammation.

**Figure 2 ijms-26-09216-f002:**
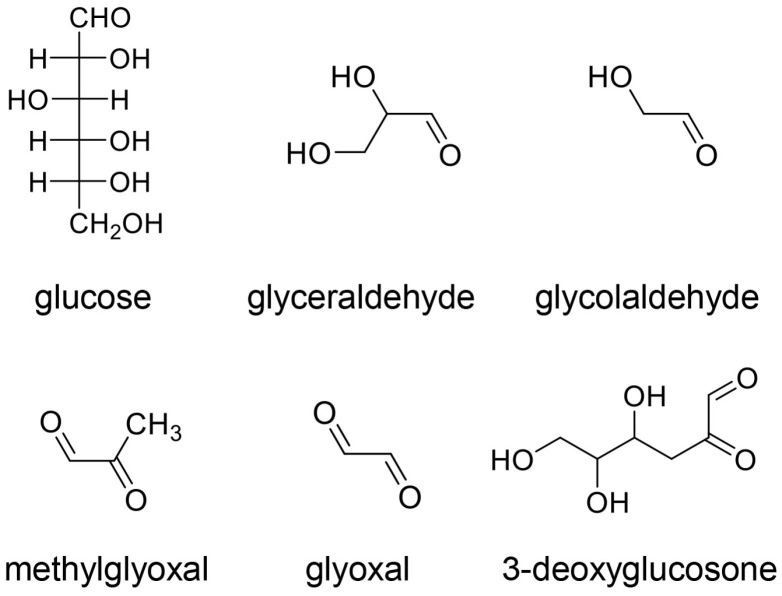
Origins of AGEs [6,9,11,12]. Glucose is a hexose, and glyceraldehyde is a triose. Glyceraldehyde, glycolaldehyde, methylglyoxal, glyoxal, and 3-deoxyglucosone can be generated from glucose via enzymatic and non-enzymatic (e.g., oxidation) reactions.

**Figure 3 ijms-26-09216-f003:**
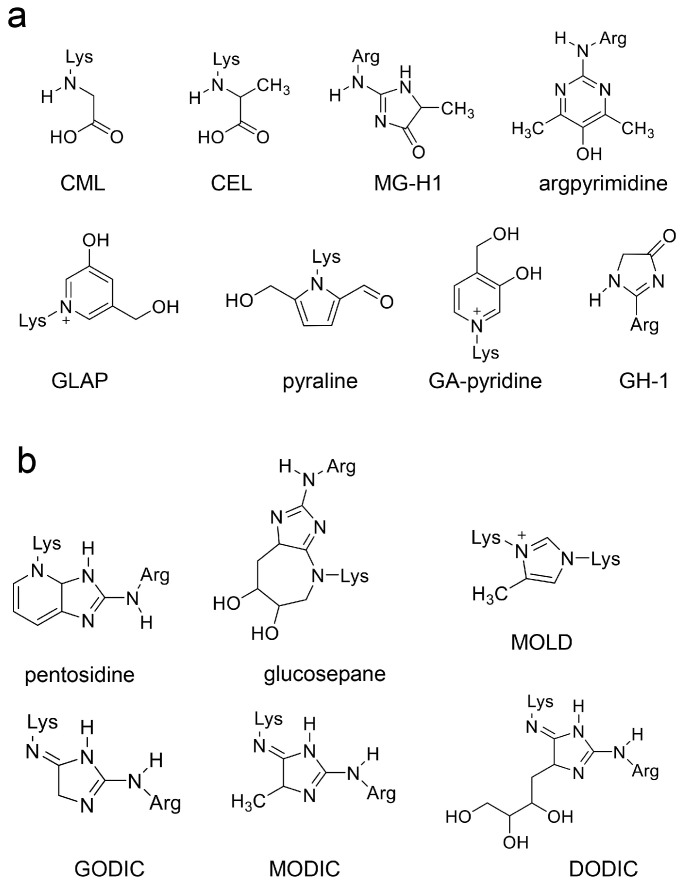
Free types of AGEs [6,9,11,12]. (**a**) AGEs containing one amino acid. CML, *N*^ε^-carboxymethyl-lysine; CEL, *N*^ε^-carboxyethyl-lysine; MG-H1, methylglyoxal-hydro-imidazolone; GLAP, glyceraldehyde-derived pyridinium; GH-1, glyoxal-hydro-imidazolone. (**b**) AGEs containing two amino acids. MOLD, methylglyoxal-derived imidazolium cross-link; GODIC, glyoxal-derived imidazolium cross-link; MODIC, methylglyoxal-derived imidazolium cross-link; DODIC, 3-deoxyglucosone-derived imidazolium cross-link.

**Figure 4 ijms-26-09216-f004:**
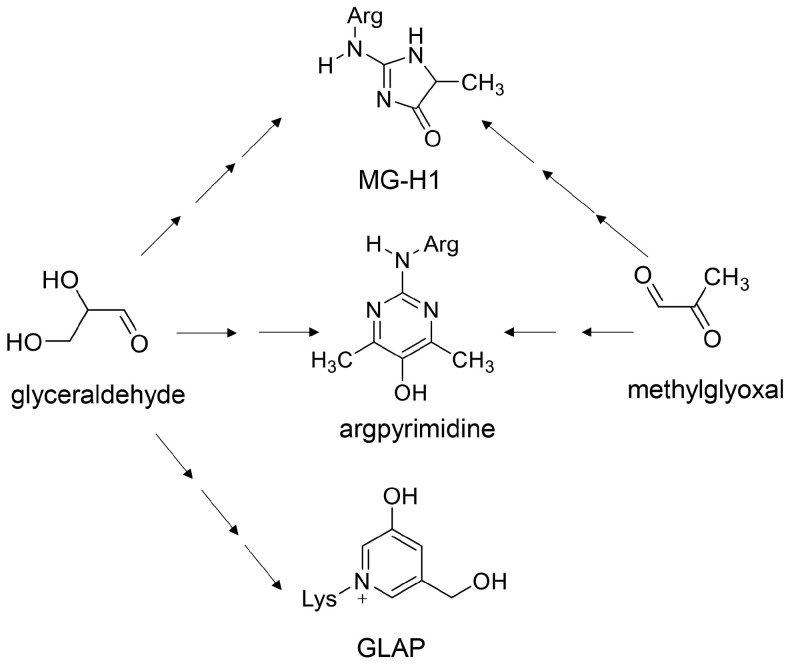
Crude AGE patterns [6,11,12]. MG-H1, argpyrimidine, and GLAP can be generated from glyceraldehyde, though MG-H1 and argpyrimidine can also be generated from methylglyoxal. Black arrow shows a step in enzymatic/non-enzymatic reactions. MG-H1, methylglyoxal-hydro-imidazolone; GLAP, glyceraldehyde-derived pyridinium.

**Figure 5 ijms-26-09216-f005:**
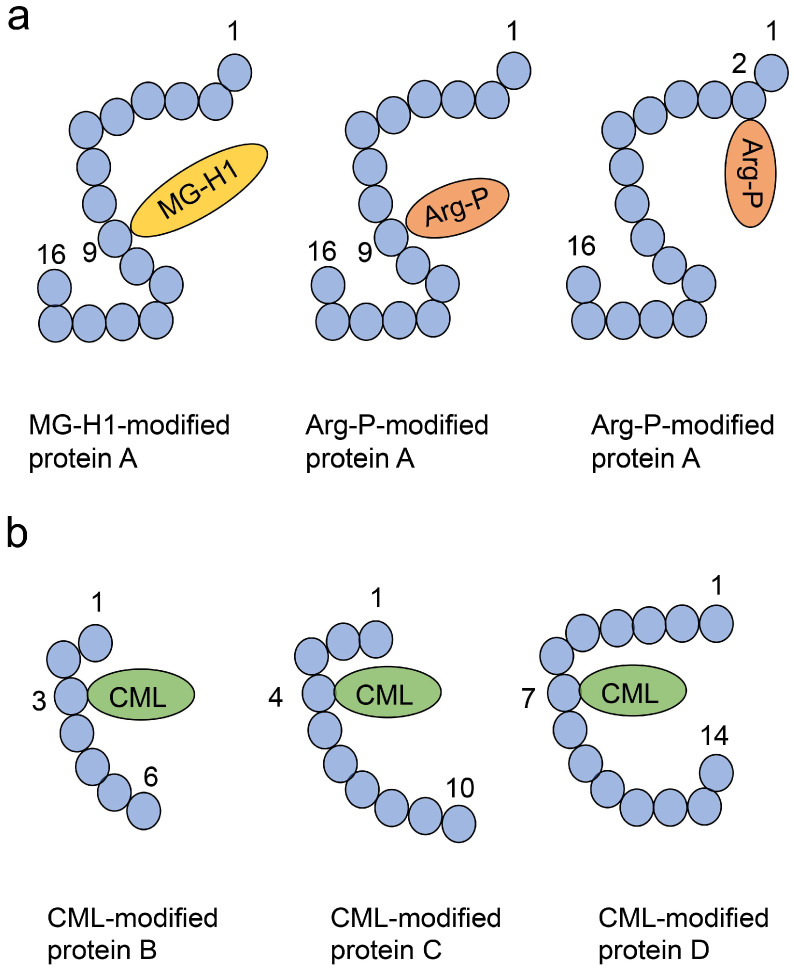
Type 1 and 2 diverse AGE patterns [11,12,88]. Closed blue circles indicate amino acids. Black number indicates the number of amino acid residues. (**a**) Type 1 diverse AGE pattern. Left and middle, middle and right, and left and right protein A structures indicate type 1A, 1B, and 1C diverse AGE patterns. MG-H1, methylglyoxal-hydro-imidazolone; Arg-P, argpyrimidine. (**b**) Type 2 diverse AGE pattern. CML; *N*^ε^-carboxymethyl-lysine.

**Figure 6 ijms-26-09216-f006:**
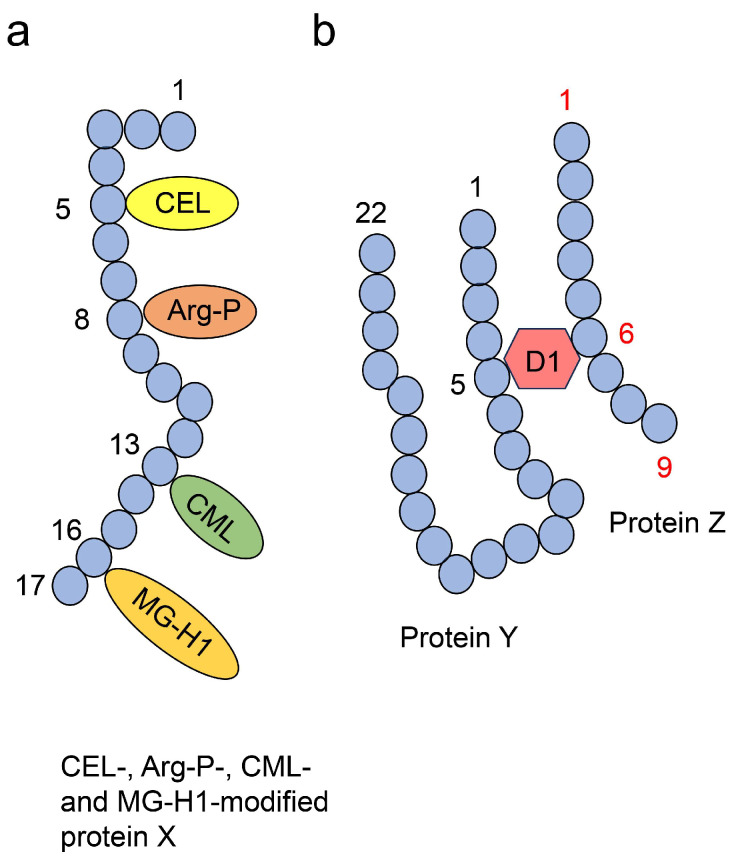
Type 1 and 2 multiple AGE patterns. Closed blue circles indicate amino acids. Black or red number indicates the number of amino acid residues [6,11,12,87,88]. (**a**) Type 1 multiple AGE pattern. CEL, *N*^ε^-carboxyethyl-lysine; Arg-P, argpyrimidine; CML, *N*^ε^-carboxymethyl-lysine; MG-H1, methylglyoxal-hydro-imidazolone. (**b**) Type 2 multiple AGE pattern. Proteins Y and Z are linked with D1—free AGE—involving two amino acids (the 5th amino acid residue in protein Y and the 6th amino acid residue in protein Z).

**Figure 7 ijms-26-09216-f007:**
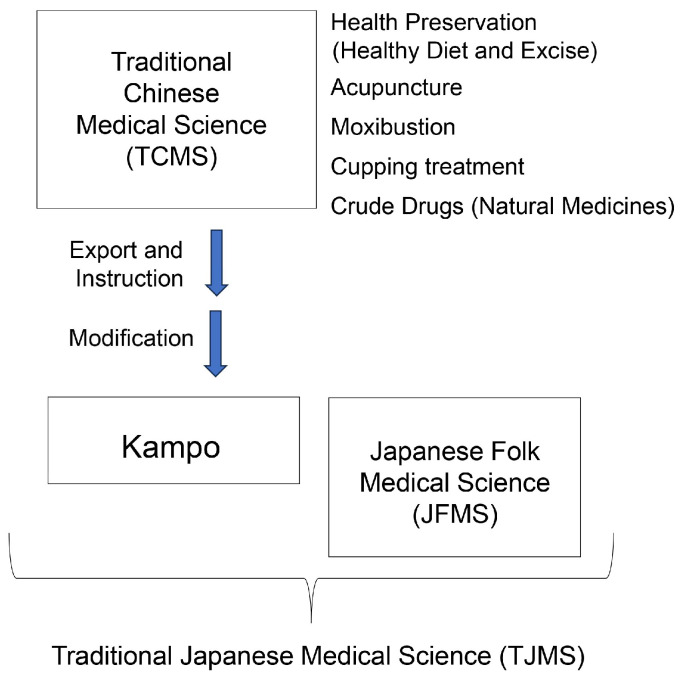
Traditional Japanese medical science (TJMS) comprises Kampo, Japanese folk medical science (JFMS), and traditional Chinese medical science (TCMS).

**Figure 8 ijms-26-09216-f008:**
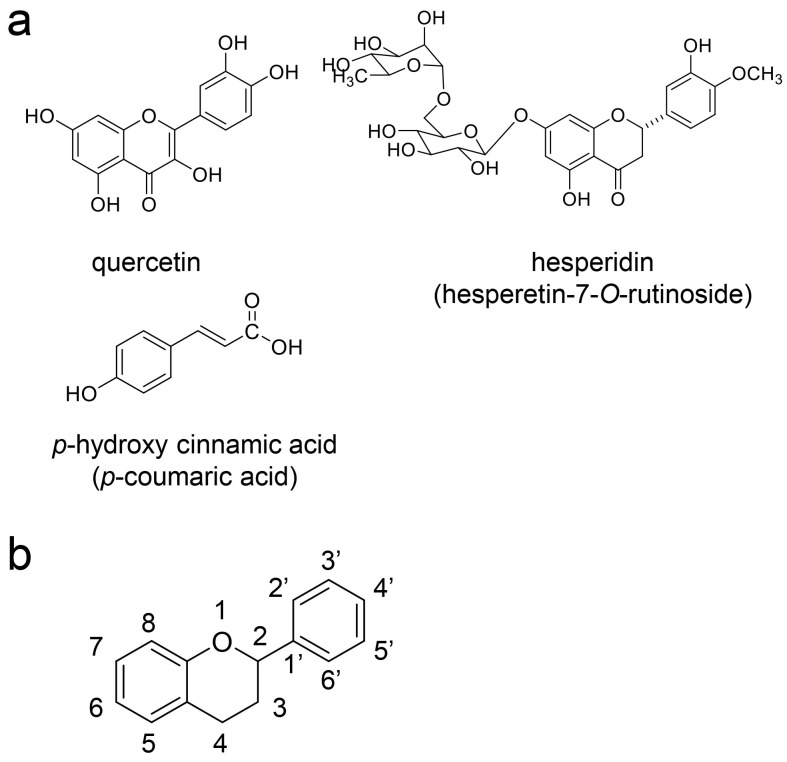
Chemical structures of study molecules. (**a**) Quercetin, hesperidin, and *p*-coumaric acid. (**b**) The structure of the flavonoid skeleton. Numbers indicate the location of the functional group (e.g., hydroxyl, carbonyl).

**Figure 9 ijms-26-09216-f009:**
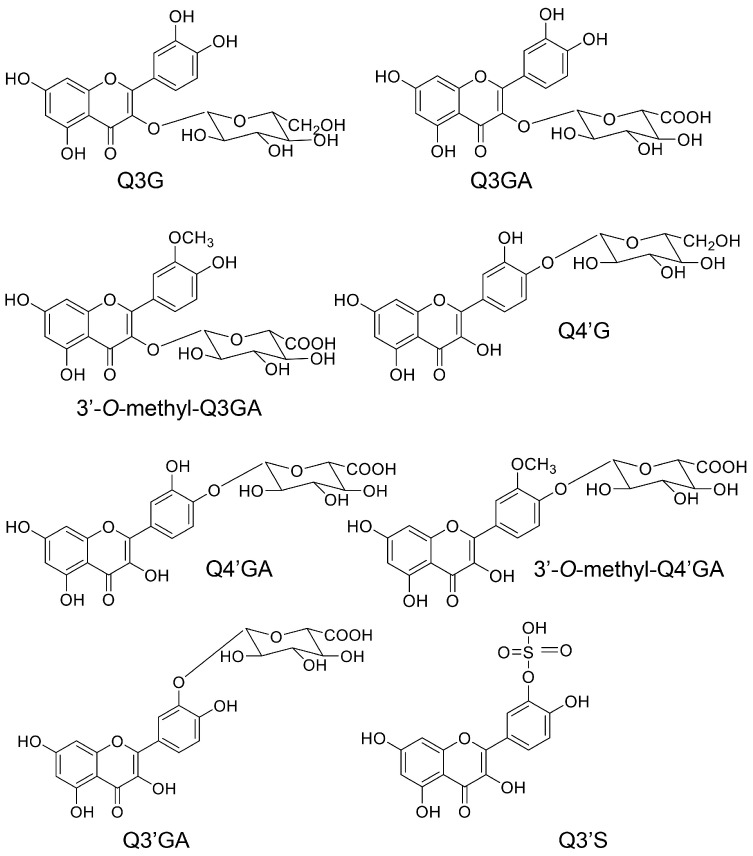
Metabolites of quercetin in the blood [129,130,131]. Q3G, quercetin-3-*O*-glucoside; Q3GA, quercetin 3-*O*-glucuronide; 3′-*O*-methyl-Q3GA, 3′-*O*-methylquercetin-*O*-glucuronide; Q4′G, quercetin-4′-*O*-glucoside; Q4′GA, quercetin-4′-*O*-glucuronide; 3′-O-methyl-Q4′GA, 3′-*O*-methylquercetin-4-*O*-glucuronide; Q3′GA, quercetin-3′-glucuronide; Q3′S, quercetin-3′-*O*-sulfate.

**Figure 10 ijms-26-09216-f010:**
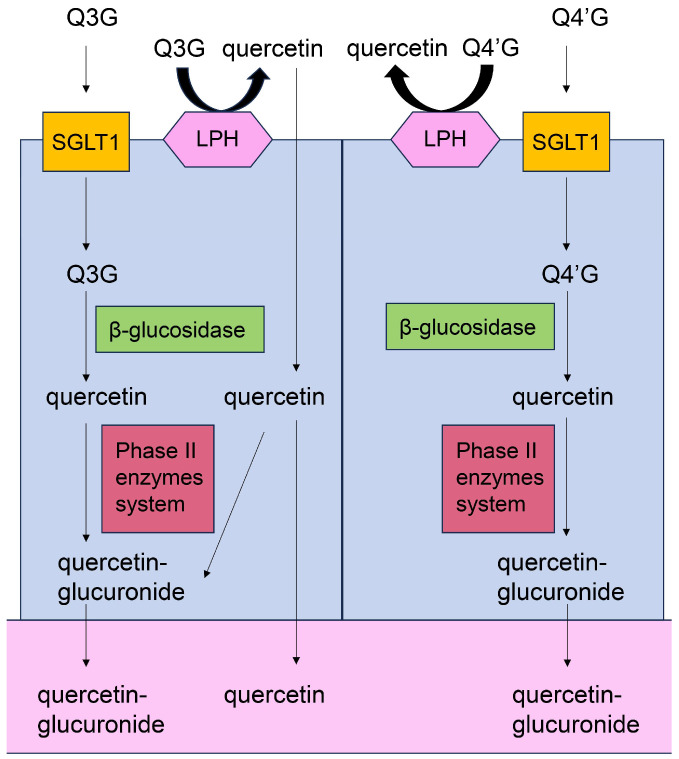
Model of Q3G, Q4′G, and quercetin absorption into the small intestine, and leaking of glucuronide-quercetin and quercetin into the blood or lymph vessels. Closed blue square indicates small-intestinal epithelial cells. Open pink area indicates blood or lymph vessel. Closed yellow square indicates sodium-glucose cotransporter 1 (SGLT1). Closed pink hexagon indicates lactase phlorizin hydrolase (LPH). Closed green square indicates cytosolic β-glucosidase. Closed red square indicates phase II enzyme system. Q3G, quercetin-3-*O*-glucoside; Q4′GA, quercetin-4′-*O*-glucuronide.

**Figure 11 ijms-26-09216-f011:**
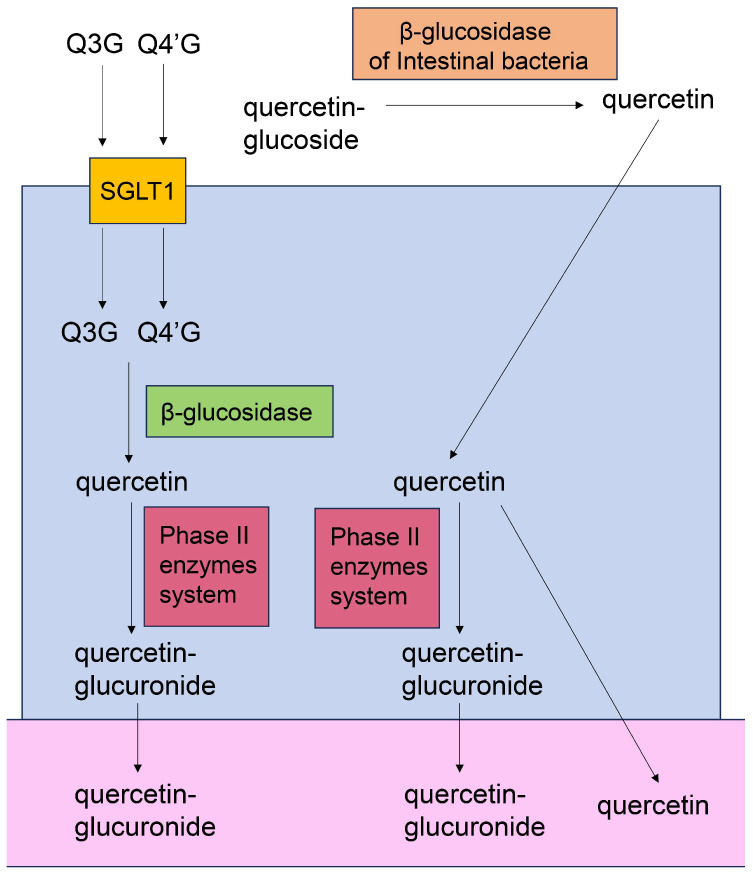
Model of Q3G, Q4′G, and quercetin absorption into the large intestine, and leaking of quercetin-glucuronide and quercetin into the blood or lymph vessels [143,144,145,146,147]. Closed blue square indicates large-intestinal epithelial cells. Open pink area indicates blood or lymph vessel. Closed orange square indicates β-glucosidase of internal bacteria. Closed yellow square indicates sodium-glucose cotransporter 1 (SGLT1). Closed green square indicates cytosolic β-glucosidase. Closed red square indicates phase II enzyme system. Q3G, quercetin-3-*O*-glucoside; Q4′GA, quercetin-4′-*O*-glucuronide.

**Figure 12 ijms-26-09216-f012:**
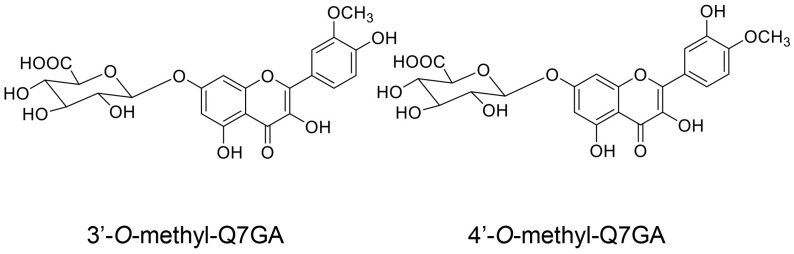
Metabolites detected in HepG2 cells treated with quercetin [147]. Q7GA, quercetin-7-*O*-glucuronide.

**Figure 13 ijms-26-09216-f013:**
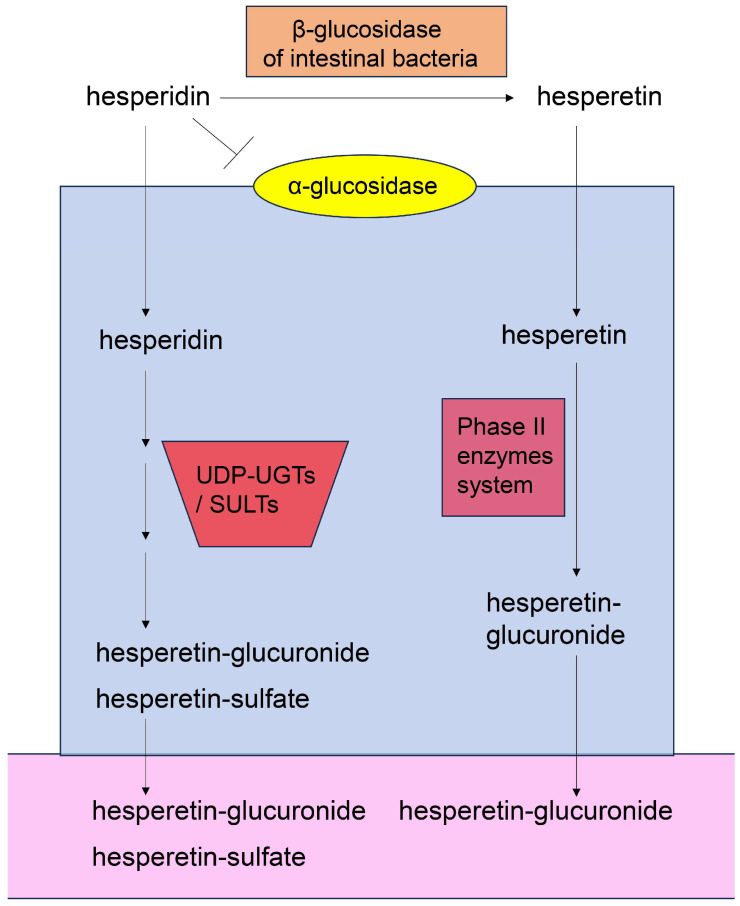
Model of the absorption of hesperidin and hesperetin in the small/large intestine, and leaking of hesperidin-glucuronide and hesperetin into the blood or lymph vessels [148,149,150,151]. Closed blue square indicates small/large intestine epithelial cells. Open pink area indicates blood or lymph vessel. Closed orange square indicates β-glucosidase of intestinal bacteria. Closed yellow circle indicates α-glucosidase. Closed red trapezoid indicates UDP-UGT/SULT. UGT, glucuronosyltransferase; SULT, sulfotransferase. Closed red square indicates phase II enzyme system.

**Figure 14 ijms-26-09216-f014:**
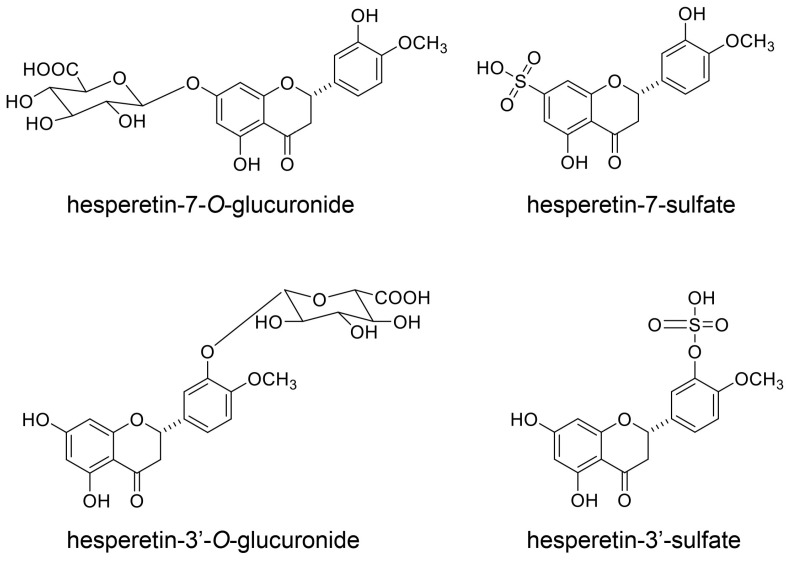
Metabolites of hesperidin detected in cytosolic fractions of small intestine, large intestine, and hepatic tissue [151,152].

**Figure 15 ijms-26-09216-f015:**
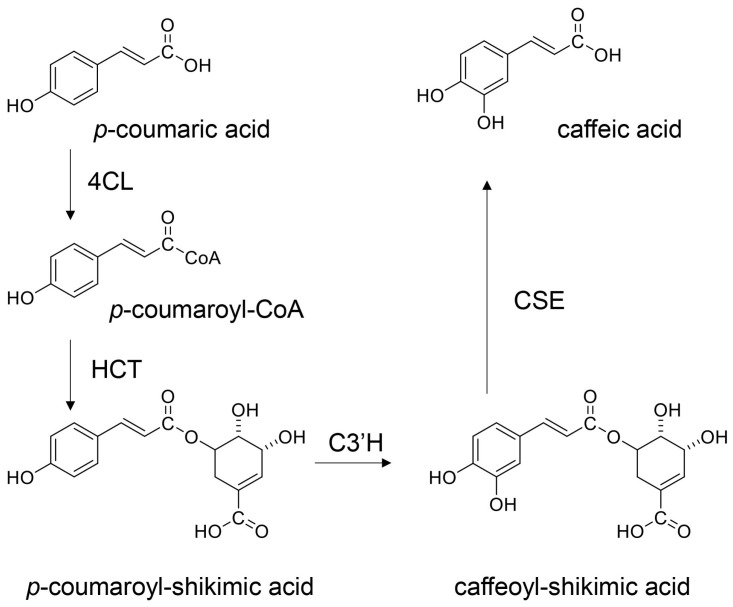
Metabolic pathway of *p*-coumaric acid in plants [158,159]. 4CL, 4-coumaric CoA-ligase; HCT, hydroxycinnamoyl transferase; C3′H, *p*-coumaroyl-shikimic acid-3-hydroxytransferase; CSE, caffeoyl shikimic esterase.

## Data Availability

The data presented in this study are available in the article.

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
