# Peer review of "Potential of Orally Administered Quercetin, Hesperidin, and p-Coumaric Acid in Suppressing Intra-/Extracellular Advanced Glycation End-Product-Induced Cytotoxicity in Proximal Tubular Epithelial Cells"

_ijms, 2025, doi:10.3390/ijms26189216_

Round 1

Reviewer 1 Report (New Reviewer)

Comments and Suggestions for Authors

This review article explores the potential of three natural compounds—quercetin, hesperidin, and p-coumaric acid, derived from Quercus salicina/Q. stenophylla leaves—in protecting proximal tubular epithelial (PTE) cells from cytotoxicity induced by advanced glycation end-products (AGEs). It summarizes how these compounds, through their parent forms and metabolites, may inhibit intracellular AGE generation via carbonyl-trapping mechanisms and attenuate extracellular AGE-RAGE/TLR4 signaling. The manuscript also reviews their absorption, metabolism, and transport following oral administration, highlighting both mechanistic insights and gaps. Overall, it provides a foundation for future preclinical and clinical studies aimed at developing natural compound-based therapies for AGE-related renal dysfunction. Here are some minor comments need to be addressed.

  1. Both “tool-like receptor 4” and “Toll-like receptor 4” appear in the text. Please standardize to “Toll-like receptor 4 (TLR4)” throughout. Correct “diabetes mellites” (line 61) to “diabetes mellitus.” In Section 5.6.5, “some molecules nay be metabolized” should be corrected to “may.”
  2. Citation formatting – Ensure uniform style. For example, in several places references are grouped inconsistently (e.g., [6,12] vs [6–12]). Please check journal style.
  3. Figures quality – Some schematic figures (e.g., Figures 5 and 6) appear low-resolution. Consider improving image quality to ensure clarity upon publication.
  4. While the review extensively discusses oral administration, absorption, and metabolism of quercetin, hesperidin, and p-coumaric acid, there is little in vivo data presented regarding their efficacy in proximal tubular epithelial (PTE) cells. Please include and critically discuss available animal studies or emphasize the need for in vivo confirmation.
  5. In this manuscript, most of the evidence is drawn from in vitro systems. Please clarify which findings are supported by clinical or preclinical studies versus cell-based assays to avoid overgeneralization.

Author Response

Response Letter to Reviewers’ Comments

Responses to Reviewer 1

Dear Reviewer 1:

Thank you for giving us the opportunity to submit a revised draft of our manuscript titled “Potential of Orally Administered Quercetin, Hesperidin, and p-Coumaric Acid in Suppressing Intra-/Extracellular Advanced Glycation End-Product-Induced Cytotoxicity in Proximal Tubular Epithelial Cells” to the International Journal of Molecular Sciences (manuscript ID: ijms-3834784). We appreciate the time and effort you have taken to provide valuable feedback on our manuscript; your comments have enriched the manuscript and helped us produce a more balanced account of our research. The manuscript has been reviewed by a professional English editor (Editage) to address all grammatical and syntax errors and improve the overall readability of the document.

We changed the title. The previous title: “Potential of Orally Administrated Quercetin, Hesperidin, and p-Coumaric Acid in Suppressing Intra-/Extracellular Advanced Glycation End-Product-Induced Cytotoxicity in Proximal Tubular Epithelial Cells”.

Administrated” was changed to “Administered”.

We inserted a new reference 3 (PMID: 39408078) and removed the previous reference 3.

We revised Figures 3, 5, 6, 9, 10, 11, and 13 to modify the dpi for publication (redesigned these at 450 dpi).

We revised Figure 7 to improve readability from the upper to lower regions.

We inserted a new Section 5.6 in the revised manuscript and removed Section 5.6.1 from the previous version.

Comments and Suggestions for Authors

This review article explores the potential of three natural compounds—quercetin, hesperidin, and p-coumaric acid, derived from Quercus salicina/Q. stenophylla leaves—in protecting proximal tubular epithelial (PTE) cells from cytotoxicity induced by advanced glycation end-products (AGEs). It summarizes how these compounds, through their parent forms and metabolites, may inhibit intracellular AGE generation via carbonyl-trapping mechanisms and attenuate extracellular AGE-RAGE/TLR4 signaling. The manuscript also reviews their absorption, metabolism, and transport following oral administration, highlighting both mechanistic insights and gaps. Overall, it provides a foundation for future preclinical and clinical studies aimed at developing natural compound-based therapies for AGE-related renal dysfunction. Here are some minor comments need to be addressed.

Comment 1 : Both “tool-like receptor 4” and “Toll-like receptor 4” appear in the text. Please standardize to “Toll-like receptor 4 (TLR4)” throughout. Correct “diabetes mellites” (line 61) to “diabetes mellitus.” In Section 5.6.5, “some molecules nay be metabolized” should be corrected to “may.”

Response 1: We corrected “Toll-like receptor 4” and “diabetes mellitus” in the Introduction section. We also corrected the use of “may” in the sentences in the new Section 5.7.4 (previously Section 5.6.5).

Comment 2 : Citation formatting – Ensure uniform style. For example, in several places references are grouped inconsistently (e.g., [6,12] vs [6–12]). Please check journal style.

Response 2: We checked the journal guidelines for article descriptions and included the number of references.

For the description of the Consecutive (Sequential) of references, we describe “[6–12]” in our manuscript when we submit. However, these references number will be described as “[6,7,8,9,10,11,12] in the article which is published with Web site of IJMS by Editorial Office.

More, this description of references will be change to “[6–12]” in the PDF file version which is able to be downloaded from the Web site of IJMS.

Comments 3 : Figures quality – Some schematic figures (e.g., Figures 5 and 6) appear low-resolution. Consider improving image quality to ensure clarity upon publication.

Response 3: We revised Figures 3, 5, 6, 9, 10, 11, and 13 to increase their dpi; the revised figures were designed at 450 dpi.

Moreover, we revised Figure 7 to make the information easier for the reviewer to read from top to bottom.

Comments 4 : While the review extensively discusses oral administration, absorption, and metabolism of quercetin, hesperidin, and p-coumaric acid, there is little in vivo data presented regarding their efficacy in proximal tubular epithelial (PTE) cells. Please include and critically discuss available animal studies or emphasize the need for in vivo confirmation.

Response 4: We inserted a new Section 5.6, in which we explained that (i) there are no reports that quercetin, hesperidin, and p-coumaric acid inhibit the generation of intracellular AGEs or suppressed cytotoxicity of extracellular AGE-RAGE/TLR4 signaling in PTE cells in vitro, in vivo (animal model studies), or in patients, (ii) these three compounds should be analyzed following oral administration to confirm their anti-AGE effects in PTE cells in vivo, and (iii) further research is needed to determine whether metabolites of these compounds exert anti-AGE effects in PTE cells, as orally administered compounds are absorbed and metabolized.

We described this information in the new Section 5.6.

Moreover, we removed Section 5.6.1 from our previous manuscript.

Comments 5 : In this manuscript, most of the evidence is drawn from in vitro systems. Please clarify which findings are supported by clinical or preclinical studies versus cell-based assays to avoid overgeneralization.

Response 5: We inserted “in vitro” and “in vivo” into the sentences describing how quercetin, hesperidin, and p-coumaric acid inhibited the generation of intracellular AGEs and suppressed extracellular AGE-RAGE/TLR4 signaling in Sections 5.4 and 5.5.

Moreover, we clarified that information on clinical investigations was not included in Sections 5.4 and 5.5.

Reviewer 2 Report (New Reviewer)

Comments and Suggestions for Authors

The manuscript entitled "Potential of Orally Administrated Quercetin, Hesperidin, and p-Coumaric acid in Suppressing Intra-/Extracellular Advanced Glycation End-Product-Induced Cytotoxicity in Proximal Tubular Epithelial Cells" brings important information about the AGE products as a diabetes mellitus complication. 

Observation

Please insert a chapter in Introduction about the pathogenesis of urinary stone formation in diabetes mellitus. 

Author Response

Response Letter to Reviewers’ Comments

Responses to Reviewer 2

Dear Reviewer 2:

Thank you for giving us the opportunity to submit a revised draft of our manuscript titled “Potential of Orally Administered Quercetin, Hesperidin, and p-Coumaric Acid in Suppressing Intra-/Extracellular Advanced Glycation End-Product-Induced Cytotoxicity in Proximal Tubular Epithelial Cells” to the International Journal of Molecular Sciences (manuscript ID: ijms-3834784). We appreciate the time and effort you have taken to provide valuable feedback on our manuscript; your comments have enriched the manuscript and helped us produce a more balanced account of our research. The manuscript has been reviewed by a professional English editor (Editage) to address all grammatical and syntax errors and improve the overall readability of the document.

We changed the title. The previous title: “Potential of Orally Administrated Quercetin, Hesperidin, and p-Coumaric Acid in Suppressing Intra-/Extracellular Advanced Glycation End-Product-Induced Cytotoxicity in Proximal Tubular Epithelial Cells”.

Administrated” was changed to “Administered”.

We inserted a new reference 3 (PMID: 39408078) and removed the previous reference 3.

We revised Figures 3, 5, 6, 9, 10, 11, and 13 to modify the dpi for publication (redesigned at 450 dpi).

We revised Figure 7 to improve readability from the upper to lower regions.

We inserted a new Section 5.6 in the revised manuscript and removed Section 5.6.1 from the previous version.

Comments and Suggestions for Authors

The manuscript entitled "Potential of Orally Administrated Quercetin, Hesperidin, and p-Coumaric acid in Suppressing Intra-/Extracellular Advanced Glycation End-Product-Induced Cytotoxicity in Proximal Tubular Epithelial Cells" brings important information about the AGE products as a diabetes mellitus complication.

Observation

Comment 1: Please insert a chapter in Introduction about the pathogenesis of urinary stone formation in diabetes mellitus.

Response 1: We described in the Introduction section that some researchers have reported that diabetes mellitus induces and promotes the production of urinary stones.

Furthermore, in Section 2.1, we inserted information that sodium-glucose cotransporter-2 (SGLT2) inhibitors reduce the production of urinary stones, and that some researchers have revealed the mechanisms by which diabetes mellitus induces and promotes urinary stone formation (highlighted in yellow).

To explain this information, we added a new reference 3 in our revised manuscript.

Reviewer 3 Report (New Reviewer)

Comments and Suggestions for Authors

International Journal of Molecular Sciences (Manuscript ID: ijms-3834784), Comments to the Authors:

Title: Potential of Orally Administrated Quercetin, Hesperidin, and p-Coumaric acid in Suppressing Intra-/Extracellular Advanced Glycation End-Product-Induced Cytotoxicity in Proximal Tubular Epithelial Cells

Comments

The submitted review focused on investigating the effects of quercetin, hesperidin, and p-coumaric acid on PTE cells in terms of their metabolism following oral administration and the associated organs and bacteria. Current evidence indicates that, in PTE cells, non-metabolized quercetin and p-coumaric acid may suppress intra-/extracellular AGE-induced cytotoxicity, while the metabolites of quercetin and hesperidin may inhibit the generation of AGEs. However, little is known of the effects of p-coumaric acid metabolites. Quercetin, hesperidin, and p-coumaric acid may collectively suppress the cytotoxicity of intra-/extracellular AGEs in PTE cells.

I think the submitted review can be accepted after the authors respond to the following comments:

  1. The entire review is built around the authors' hypothesis that quercetin, hesperidin, and p-coumaric acid can suppress AGE-induced cytotoxicity in PTE cells, but this is largely speculative and not well-supported by direct experimental evidence. The authors should expand the Introduction and add a dedicated section reviewing any PTE/HK-2 cell studies on these compounds. If none exist, the authors should clearly frame the entire review as hypothetical and discuss why PTE cells might respond similarly to other models, with caveats. The authors should quantify the evidence gap.
  2. The authors should provide a detailed section to pharmacokinetic data. The authors should include tables comparing plasma/urine levels post-oral dosing in humans/animals, metabolite profiles, and renal-specific transport. The authors should discuss how gut bacteria might influence efficacy.
  3. Certain concepts, such as the role of AGEs in PTE cell dysfunction and the general mechanisms of action of the compounds (carbonyl trapping, RAGE/TLR4 inhibition), are repeated across different sections. This makes the review less concise and can be distracting for the reader.
  4. The authors should provide a detailed mechanistic insight into how these compounds or their metabolites specifically interact with the stated pathways at a molecular level within PTE cells. The authors should highlight what are the specific enzymes or receptors involved in the carbonyl trapping by these compounds? How do they modulate RAGE/TLR4 signaling beyond general inhibition?
  5. The authors introduced new AGE categorization systems (crude, diverse, multiple patterns) without sufficient validation or acceptance in the field. Can the authors provide detailed explanation on why they introduced these classifications.
  6. The authors stated that Urocalun and Choreito as urinary stone treatments but provided no comparative data on efficacy vs. modern drugs or other anti-AGE agents. The authors should add a section comparing these compounds to approved therapies. The authors should discuss limitations of folk medicine compared to modern drug therapy.

Author Response

Response Letter to Reviewers’ Comments

Responses to Reviewer 3

Dear Reviewer 3:

Thank you for giving us the opportunity to submit a revised draft of our manuscript titled “Potential of Orally Administered Quercetin, Hesperidin, and p-Coumaric Acid in Suppressing Intra-/Extracellular Advanced Glycation End-Product-Induced Cytotoxicity in Proximal Tubular Epithelial Cells” to the International Journal of Molecular Sciences (manuscript ID: ijms-3834784). We appreciate the time and effort you have taken to provide valuable feedback on our manuscript; your comments have enriched the manuscript and helped us produce a more balanced account of our research. The manuscript has been reviewed by a professional English editor (Editage) to address all grammatical and syntax errors and improve the overall readability of the document.

We changed the title. The previous title: “Potential of Orally Administrated Quercetin, Hesperidin, and p-Coumaric Acid in Suppressing Intra-/Extracellular Advanced Glycation End-Product-Induced Cytotoxicity in Proximal Tubular Epithelial Cells”.

Administrated” was changed to “Administered”.

We inserted a new reference 3 (PMID: 39408078) and removed the previous reference 3.

We revised Figures 3, 5, 6, 9, 10, 11, and 13 to modify the dpi for publication (redesigned at 450 dpi).

We revised Figure 7 to improve readability from the upper to lower regions.

We inserted a new Section 5.6 in the revised manuscript and removed Section 5.6.1 from the previous version.

Comments

The submitted review focused on investigating the effects of quercetin, hesperidin, and p-coumaric acid on PTE cells in terms of their metabolism following oral administration and the associated organs and bacteria. Current evidence indicates that, in PTE cells, non-metabolized quercetin and p-coumaric acid may suppress intra-/extracellular AGE-induced cytotoxicity, while the metabolites of quercetin and hesperidin may inhibit the generation of AGEs. However, little is known of the effects of p-coumaric acid metabolites. Quercetin, hesperidin, and p-coumaric acid may collectively suppress the cytotoxicity of intra-/extracellular AGEs in PTE cells.

I think the submitted review can be accepted after the authors respond to the following comments:

Comment 1 : The entire review is built around the authors' hypothesis that quercetin, hesperidin, and p-coumaric acid can suppress AGE-induced cytotoxicity in PTE cells, but this is largely speculative and not well-supported by direct experimental evidence. The authors should expand the Introduction and add a dedicated section reviewing any PTE/HK-2 cell studies on these compounds. If none exist, the authors should clearly frame the entire review as hypothetical and discuss why PTE cells might respond similarly to other models, with caveats. The authors should quantify the evidence gap.

Response 1: We inserted the following sentence in the Introduction section:

“To test this hypothesis, we predicted and considered various conditions (see Sections 5.6 and 5.7).”

Moreover, we added a new Section 5.6 in our revised manuscript and removed Section 5.6.1 from the previous version.

Comment 2 : The authors should provide a detailed section to pharmacokinetic data. The authors should include tables comparing plasma/urine levels post-oral dosing in humans/animals, metabolite profiles, and renal-specific transport. The authors should discuss how gut bacteria might influence efficacy.

Response 2: Because we could not provide a detailed section on pharmacokinetic data, we described this fact and their potential benefits for future investigations in the Limitations section.

Comment 3 : Certain concepts, such as the role of AGEs in PTE cell dysfunction and the general mechanisms of action of the compounds (carbonyl trapping, RAGE/TLR4 inhibition), are repeated across different sections. This makes the review less concise and can be distracting for the reader.

Response 3: We revised Sections 5.4 and 5.5 and inserted a new Section 5.6.

In Section 5.4, we introduced quercetin, hesperidin, and p-coumaric acid inhibited the generation of intracellular AGEs in cells other than PTE cells.

In Section 5.5, we introduced that quercetin, hesperidin, and p-coumaric acid suppressed extracellular AGEs-RAGE/TLR4 signaling in cells other than PTE cells.

In Section 5.6, we hypothesized that these compounds can inhibit the generation of intracellular AGEs and suppress extracellular AGEs-RAGE/TLR4 signaling in PTE cells, and we presented the issues that need to be addressed to prove this hypothesis.

Comment 4 : The authors should provide a detailed mechanistic insight into how these compounds or their metabolites specifically interact with the stated pathways at a molecular level within PTE cells. The authors should highlight what are the specific enzymes or receptors involved in the carbonyl trapping by these compounds? How do they modulate RAGE/TLR4 signaling beyond general inhibition?

Response 4: We revised Section 5.4 to explain that the carbonyl trap system can function as a simple organic reaction (this reaction does not require enzymes). Ketone and aldehyde groups of compounds are targets for the carbonyl trap system. We described this information in Section 5.4.

Moreover, we inserted a new Section 5.6 to explain the possibility that quercetin, hesperidin, and p-coumaric acid may modulate RAGE/TLR4 signaling in PTE cells as well as in other cells. The reasons are that (i) RAGE and TLR4 are expressed and (ii) their downstream pathways involve NF-κβ signaling in PTE cells, as in other cells.

Finally, we inserted some sentences in Section 5.7.5 to describe the possibility that orally administered quercetin and p-coumaric acid may inhibit the generation of intracellular AGEs in PTE cells via the carbonyl trap system which can act regardless of cell type.

Comment 5 : The authors introduced new AGE categorization systems (crude, diverse, multiple patterns) without sufficient validation or acceptance in the field. Can the authors provide detailed explanation on why they introduced these classifications.

Response 5: The terms “Crude AGE pattern,” “Diverse AGE pattern,” and “Multiple AGE pattern” were coined by us and are not currently accepted in the AGE research field. However, the phenomena whereby (i) various free types of AGEs (e.g., MG-H1, argpyrimidine, GLAP) are generated from a single compound (e.g., glyceraldehyde) in vitro (cultured cells) and in tube, and (ii) various free types of AGEs modify a single protein or molecule in vitro and in vivo, have been reported by some researchers using ESI-MS and MALDI-MS. Although these phenomena have been described, and this understanding may be important for future AGE research, they have not yet been formally named or categorized.

Therefore, we introduced our categorization in this review article, although it has not been accepted in the AGE research field.

We described this information and explanation in Sections 3.2.3, 3.2.4, and 3.2.5.

Comment 6 : The authors stated that Urocalun and Choreito as urinary stone treatments but provided no comparative data on efficacy vs. modern drugs or other anti-AGE agents. The authors should add a section comparing these compounds to approved therapies. The authors should discuss limitations of folk medicine compared to modern drug therapy.

Response 6: In Japan, there is no medicine aimed at anti-intra-/extracellular AGE effects that is available for clinical use (hospital, clinic, or pharmacy) in 2025. However, many researchers have attempted to analyze existing medicine that may show anti-intra-/extracellular AGE effects.

When we surveyed the PubMed database for references on the relationship between medicines for urinary stones and AGEs for this review article, we were unable to obtain relevant information (as of August–September 2025).

In this review article, we hypothesized the potential of natural compounds in medicines selected for the treatment and/or prevention of urinary stones. Therefore, we did not focus on other modern European or Traditional medicines (e.g., anti-cancer medicines, anti-inflammatory medicines for oral squamous cell carcinoma).

Moreover, we revised Section 5.2 to describe the reason we focused on Q. salicina/Q. stenophylla leaf extract as part of Japanese folk medicine.

Finaly, we described the issue of analyzing and investigating Q. salicina/Q. stenophylla leaf extract in comparison with European modern medicines in the Limitations section.

Round 2

Reviewer 3 Report (New Reviewer)

Comments and Suggestions for Authors

After reading the authors response to my comments I think the revised review can be accepted for publication 

This manuscript is a resubmission of an earlier submission. The following is a list of the peer review reports and author responses from that submission.

Round 1

Reviewer 1 Report

Comments and Suggestions for Authors

This review article explores the potential anti-glycation effects of natural compounds found in Urocalun, a traditional Japanese herbal medicine derived from Quercus salicina or Q. stenophylla, focusing on their ability to suppress intra- and extracellular AGE-induced cytotoxicity in proximal tubular epithelial (PTE) cells. The authors specifically focus on quercetin, hesperidin, and p-coumaric acid as candidate compounds contributing to these effects.

While the manuscript discusses the potential anti-AGE effects of Urocalun constituents, it has several fundamental issues that significantly undermine its scientific rigor and reliability. Based on the concerns outlined below, I do not recommend this manuscript for publication in its current form. 

1. Disconnect between title and content 
−    The title refers to Urocalun as a multi-component traditional medicine and does not indicate a focus on individual purified compounds. However, the manuscript centers almost exclusively on three specific molecules: quercetin, hesperidin, and p-coumaric acid. This focus is not clearly justified and creates disconnect between the title and the actual content. Given that Urocalun is a crude botanical formulation rather than a single-compound drug, the review would be more appropriate if it addressed the extract as a whole or discussed a broader range of its phytochemicals. Selecting only three compounds without a clear rationale leads to conceptual imbalance. 

2. Lack of clear evidence for compound inclusion 
−    The cited literature (references 29 to 31) confirms the presence of quercetin and hesperidin but does not provide direct evidence for the inclusion of p-coumaric acid in Urocalun. For instance, Reference [30] lists ferulic acid and p-hydroxycinnamic acid but not p-coumaric acid among the identified constituents.
−    Furthermore, a 2020 review titled ‘Bioactive compounds and antioxidant activities of Quercus salicina Blume extract’ documents a wide variety of phenolic acids (such as gallic, chlorogenic, caffeic, ellagic) and flavonoids (such as rutin, naringin, kaempferol, myricetin), many of which are more prominently featured in the plant's composition than p-coumaric acid. These more abundant or relevant bioactive compounds are not addressed in the current manuscript. 

3. Absence of quantitative compositional data 
−    The manuscript lacks any quantitative information on the concentration or proportion of the three selected compounds in Urocalun. There is also no indication of whether these compounds are present at pharmacologically relevant levels for AGE inhibition. Without such data, the mechanistic discussion remains speculative and unsupported. 

4. Risk of misleading therapeutic implications 
−    The manuscript gives the impression that Urocalun is effective against AGE-related cytotoxicity based on indirect and limited evidence. This could lead readers to interpret the review as endorsing therapeutic use without sufficient validation. Given the commercial interest in natural products, such implications are ethically problematic and may promote premature or inappropriate application.

Author Response

Response Letter to Reviewers’ Comments

Responses to Reviewer 1

Dear Reviewer 1:

Thank you for providing us with the opportunity to revise our manuscript titled “Potential of Orally Administrated Quercetin, Hesperidin, and p-Coumaric acid in Suppressing Intra-/Extracellular Advanced Glycation End-Product-Induced Cytotoxicity in Proximal Tubular Epithelial Cells” for submission to the International Journal of Molecular Sciences (manuscript ID: ijms-3714187). We appreciate the time and effort you have taken to provide valuable feedback on our manuscript; your comments have enriched the manuscript and helped us produce a more balanced account of our research. The manuscript has been reviewed by a professional English language editor (Editage) to address all grammatical and syntax errors and improve the overall readability of the document.

In this revised manuscript, we performed the following Major revision.

  • We changed the title (Previous title: “Potential of Natural Products in Urocalun for Suppressing Intra-/Extracellular Advanced Glycation End-Product-Induced Cytotoxicity in Proximal Tubular Epithelial Cells”).
  • We removed the information on Urocalun from main text.
  • We focused our review on quercetin, hesperidin, and p-coumaric acid, and indicate that the information is used as the basis for assessing the effects of salicina/Q. stenophylla leaf extract against intra/extracellular AGE-induced cytotoxicity in proximal tubular epithelial (PTE) cells.
  • We removed the previous ref. 32 and inserted a new reference (report described by Aung et al., PMID 32296555).
  • We rewrote the Abstract, Introduction, Discussion, Limitations, Conclusion, and Future directions (yellow-highlighted text)
  • We included an explanation for Kampo medicine, indicating that traditional Chinese medical science was transported into ancient Japan directly from the Chinese mainland and via the Korean peninsula.
  • We removed “original Japanese medicine” and “original Japanese medical science” in the previous manuscript, and described “Japanese folk medicine” in the revised manuscript, which should accurately classify salicina/Q. stenophylla leaf extract. If it was selected as a regular authalic medicine in China, Korea, Vietnam, or other areas, this information did not reach Japan during the ancient to early modern periods.

Comments and Suggestions for Authors

This review article explores the potential anti-glycation effects of natural compounds found in Urocalun, a traditional Japanese herbal medicine derived from Quercus salicina or Q. stenophylla, focusing on their ability to suppress intra- and extracellular AGE-induced cytotoxicity in proximal tubular epithelial (PTE) cells. The authors specifically focus on quercetin, hesperidin, and p-coumaric acid as candidate compounds contributing to these effects.

While the manuscript discusses the potential anti-AGE effects of Urocalun constituents, it has several fundamental issues that significantly undermine its scientific rigor and reliability. Based on the concerns outlined below, I do not recommend this manuscript for publication in its current form. 

Main points:

Response 1: Disconnect between title and content 
The title refers to Urocalun as a multi-component traditional medicine and does not indicate a focus on individual purified compounds. However, the manuscript centers almost exclusively on three specific molecules: quercetin, hesperidin, and p-coumaric acid. This focus is not clearly justified and creates disconnect between the title and the actual content. Given that Urocalun is a crude botanical formulation rather than a single-compound drug, the review would be more appropriate if it addressed the extract as a whole or discussed a broader range of its phytochemicals. Selecting only three compounds without a clear rationale leads to conceptual imbalance.

Response 1: We changed the title to “Potential of Orally Administrated Quercetin, Hesperidin, and p-Coumaric acid in Suppressing Intra-/Extracellular Advanced Glycation End-Product-Induced Cytotoxicity in Proximal Tubular Epithelial Cells”. We removed the information about Urocalun from the main objectives and text, focusing instead on quercetin, hesperidin, and p-coumaric acid. The revised text explains that the information for these components are based on the available investigations on Q. salicina/Q. stenophylla leaf extract in the treatment of anti-intra/extracellular AGEs-induced cytotoxicity in PTE cells.

We changed the title of Section 5.1. and explained the relationship between Urocalun and the Q. salicina/Q. stenophylla leaf extract in Japanese folk medicine.

We inserted a new Section 5.2. (Title: Limited Data on the Effects of Q. salicina/Q. stenophylla Leaf Extract Components).

As seen below, we explained our reasoning for selecting quercetin, hesperidin, and p-coumaric acid from the various natural compounds in Q. salicina/Q. stenophylla leaf extract in Section 5.3, and introduced ref. 32 (Aung et al., PMID 32296555):

“Although various flavonoids (e.g., rutin, naringin, quercetin, hesperidin, and myricetin) and phenolic acids (e.g., gallic acid, pyrogallol, caffeic acid, p-coumaric acid, and m-coumaric acid) have been isolated [32], we selected quercetin and hesperidin as typical examples of aglycon- and glycoside-type flavonoids, and p-coumaric acid as an example of phenolic acid. Although Aung et al. [32] characterized the leaf contents (μg/g) of various natural compounds such as flavonoids and phenolic acids, this study focused only on structure.”

Comment 2: Lack of clear evidence for compound inclusion 
−    The cited literature (references 29 to 31) confirms the presence of quercetin and hesperidin but does not provide direct evidence for the inclusion of p-coumaric acid in Urocalun. For instance, Reference [30] lists ferulic acid and p-hydroxycinnamic acid but not p-coumaric acid among the identified constituents.
−    Furthermore, a 2020 review titled ‘Bioactive compounds and antioxidant activities of Quercus salicina Blume extract’ documents a wide variety of phenolic acids (such as gallic, chlorogenic, caffeic, ellagic) and flavonoids (such as rutin, naringin, kaempferol, myricetin), many of which are more prominently featured in the plant's composition than p-coumaric acid. These more abundant or relevant bioactive compounds are not addressed in the current manuscript. 

Response 2: To explain that p-coumaric acid was detected and isolated from the leaves of Q. salicina/Q. stenophylla, we introduced ref. 32 (Aung et al., PMID 32296555) in Section 5.3. along with the following text:

“Although various flavonoids (e.g., rutin, naringin, quercetin, hesperidin, and myricetin) and phenolic acids (e.g., gallic acid, pyrogallol, caffeic acid, p-coumaric acid, and m-coumaric acid) have been isolated [32], we selected quercetin and hesperidin as typical examples of aglycon- and glycoside-type flavonoids, and p-coumaric acid as an example of phenolic acid. Although Aung et al. [32] characterized the leaf contents (μg/g) of various natural compounds such as flavonoids and phenolic acids, this study focused only on structure.”

Comment 3: Absence of quantitative compositional data 
−    The manuscript lacks any quantitative information on the concentration or proportion of the three selected compounds in Urocalun. There is also no indication of whether these compounds are present at pharmacologically relevant levels for AGE inhibition. Without such data, the mechanistic discussion remains speculative and unsupported. 

Response 3: We removed the information about Uroclaun from the main objectives and text. Since the scope of the review is focused on structural research, our revised manuscript lacks any quantitative information on the concentration or proportion of quercetin, hesperidin, and p-coumaric acid (non-metabolized compounds and their metabolites).

We described this information in Section 5.3.

In Limitation (Section 6.), we noted that there is no indication of whether quercetin, hesperidin, p-coumaric acid, and their metabolites occur at pharmacologically relevant levels for mitigating intra-/extracellular AGE-induced cytotoxicity in PTE cells. Although this information is useful for describing the bioactivities of molecular compounds, the aim of this study was to describe the potential transport of non-metabolized quercetin, hesperidin, and p-coumaric acid and their metabolites into PTE cells.

We described the possible effects of non-metabolized/metabolized quercetin, hesperidin, and p-coumaric acid on intra-/extracellular AGE-induced cytotoxicity in PTE cells in Sections 5.6.5., 5.6.6., and Conclusions (Section 7.).

Comment 4: Risk of misleading therapeutic implications 
−    The manuscript gives the impression that Urocalun is effective against AGE-related cytotoxicity based on indirect and limited evidence. This could lead readers to interpret the review as endorsing therapeutic use without sufficient validation. Given the commercial interest in natural products, such implications are ethically problematic and may promote premature or inappropriate application.

Response 4: In the revised manuscript, we avoided describing the effects of Urocalun or the leaves of Q. salicina/Q. stenophylla on intra-/extracellular AGE-induced cytotoxicity in PTE cells. We propose that the information on quercetin, hesperidin, and p-coumaric acid will be beneficial for investigating the effects of Q. salicina/Q. stenophylla leaf extract in the Abstract, Introduction, and Conclusion sections. These changes should reduce the misinterpretation of therapeutic implications. We also emphasize the importance of further mechanistic research on the other natural compounds in Q. salicina/Q. stenophylla leaf extract in Future directions (Section 8.).

Reviewer 2 Report

Comments and Suggestions for Authors

This study focuses on the inhibitory effects of quercetin, naringin, and p-coumaric acid in Urocalun on the cytotoxicity induced by advanced glycation end products (AGEs) inside and outside proximal tubular epithelial cells. Given the close association between AGEs and kidney-related issues in lifestyle-related diseases, the research topic holds practical significance and has potential value for understanding the mechanisms of related diseases and developing treatment methods. However, the following issues exist:

  1. The paper mainly reviews the role of the components contained in Urocalun in inhibiting the cytotoxicity of AGEs in existing research, but it lacks depth in terms of innovative data or new discoveries, and does not highlight how the research changes our understanding of the relevant phenomena.
  2. In the discussion section, the comparison of the research results with existing studies is not sufficiently in-depth, and the uniqueness and significance of the research results are not sufficiently emphasized. For example, when discussing the role of natural products in Urocalun, there is no detailed comparison of the differences between this study and other similar studies in terms of efficacy, mechanisms, etc. While some key issues, such as the classification and mechanisms of action of AGEs, are discussed in detail, there is a lack of deeper exploration and analysis.
  3. The limitations of the study are not discussed in sufficient depth. Although the study mentions that “the scope of the study neglected some factors such as amylase in the mouth and digestive enzymes in the gut,” it does not discuss in detail the specific impact these factors may have on the study results, nor does it propose specific directions and methods for follow-up studies to address these limitations.

Author Response

Response Letter to Reviewers’ Comments

Responses to Reviewer 2

Dear Reviewer 2:

Thank you for providing us with the opportunity to revise our manuscript titled “Potential of Orally Administrated Quercetin, Hesperidin, and p-Coumaric acid in Suppressing Intra-/Extracellular Advanced Glycation End-Product-Induced Cytotoxicity in Proximal Tubular Epithelial Cells” for submission to the International Journal of Molecular Sciences (manuscript ID: ijms-3714187). We appreciate the time and effort you have taken to provide valuable feedback on our manuscript; your comments have enriched the manuscript and helped us produce a more balanced account of our research. The manuscript has been reviewed by a professional English language editor (Editage) to address all grammatical and syntax errors and improve the overall readability of the document.

In this revised manuscript, we performed the following Major revision.

  • We changed the title (Previous title: “Potential of Natural Products in Urocalun for Suppressing Intra-/Extracellular Advanced Glycation End-Product-Induced Cytotoxicity in Proximal Tubular Epithelial Cells”).
  • We removed the information on Urocalun from the main text.
  • We focused on quercetin, hesperidin, and p-coumaric acid, and indicate that the information is used as the basis for investigating the effects of salicina/Q. stenophylla leaf extract against intra/extracellular AGE-induced cytotoxicity in proximal tubular epithelial (PTE) cells.
  • We removed the previous ref. 32 and inserted a new reference (report described by Aung et al., PMID 32296555).
  • We rewrote the Abstract, Introduction, Discussion, Limitations, Conclusion, and Future directions (yellow-highlighted text)
  • We included an explanation for Kampo medicine, indicating that traditional Chinese medical science was transported into ancient Japan directly from the Chinese mainland and via the Korean peninsula.

We removed “original Japanese medicine” and “original Japanese medical science” in the previous manuscript, and described “Japanese folk medicine” in the revised manuscript, which should accurately classify Q. salicina/Q. stenophylla leaf extract. If it was selected as a regular authalic medicine in China, Korea, Vietnam, or other areas, this information did not reach Japan during the ancient to early modern periods.

Comments and Suggestions for Authors

This study focuses on the inhibitory effects of quercetin, naringin, and p-coumaric acid in Urocalun on the cytotoxicity induced by advanced glycation end products (AGEs) inside and outside proximal tubular epithelial cells. Given the close association between AGEs and kidney-related issues in lifestyle-related diseases, the research topic holds practical significance and has potential value for understanding the mechanisms of related diseases and developing treatment methods. However, the following issues exist:

Comment 1: The paper mainly reviews the role of the components contained in Urocalun in inhibiting the cytotoxicity of AGEs in existing research, but it lacks depth in terms of innovative data or new discoveries, and does not highlight how the research changes our understanding of the relevant phenomena.

Response 1: We changed the title to “Potential of Orally Administrated Quercetin, Hesperidin, and p-Coumaric acid in Suppressing Intra-/Extracellular Advanced Glycation End-Product-Induced Cytotoxicity in Proximal Tubular Epithelial Cells”. We removed the information about Urocalun from the main objectives and text, focusing instead on quercetin, hesperidin, and p-coumaric acid. The revised text explains that the information for these components are based on the available investigations on Q. salicina/Q. stenophylla leaf extract in the treatment of anti-intra/extracellular AGEs-induced cytotoxicity in PTE cells.

We changed the title of Section 5.1. and explained the relationship between Urocalun and the Q. salicina/Q. stenophylla leaf extract in Japanese folk medicine.

We inserted a new Section 5.2. (Title: Limited Data on the Effects of Q. salicina/Q. stenophylla Leaf Extract Components).

We included the following explanation in Section 5.2.: Because of the lack of studies, we characterized the effects of each compound based on the available mechanistic research. Therefore, direct treatments using the extract are unsuitable for describing the in vitro effects of the various components, whereas treatments with the respective components or their metabolites may effectively elucidate their effects. This approach is especially important given the metabolic processes that natural compounds and their metabolites are exposed to during transport to PTE cells following oral administration.

We explained our reasoning for selecting quercetin, hesperidin, and p-coumaric acid from the various natural compounds in Q. salicina/Q. stenophylla leaf extract in Section 5.3, and introduced ref. 32 (Aung et al., PMID 32296555):

We provided our rationalization on the importance of non-metabolized quercetin, hesperidin, p-coumaric acid and their metabolites with regards to assessing the effects of Q. salicina/Q. stenophylla leaf extract in PTE cells in the Abstract, Introduction, and Conclusion sections.

Comment 2: In the discussion section, the comparison of the research results with existing studies is not sufficiently in-depth, and the uniqueness and significance of the research results are not sufficiently emphasized. For example, when discussing the role of natural products in Urocalun, there is no detailed comparison of the differences between this study and other similar studies in terms of efficacy, mechanisms, etc. While some key issues, such as the classification and mechanisms of action of AGEs, are discussed in detail, there is a lack of deeper exploration and analysis.

Response 2: The revised Sections 5.4 and 5.5 emphasize the lack of in vitro and in vivo research on the suppressive effects of quercetin, hesperidin, and p-coumaric acid on intra-/extracellular AGE expression in PTE cells. We also noted that no previous reports have described the relationship between the absorption and metabolism of quercetin, hesperidin, and p-coumaric acid and inhibition of intra-/extracellular AGEs.

Nevertheless, to assess therapeutic performance, it is important to characterize the transport of the respective compounds and their metabolites into PTE cells in terms of the oral route of quercetin, hesperidin, and p-coumaric acid administration, as described in Section 5.6.1.

In Sections 5.6.5 and 5.6.6., we described the potential transport of non-metabolized and/or metabolized compounds into PTE cells and consequent suppression of intra-/extracellular AGE-induced cytotoxicity.

Comment 3: The limitations of the study are not discussed in sufficient depth. Although the study mentions that “the scope of the study neglected some factors such as amylase in the mouth and digestive enzymes in the gut,” it does not discuss in detail the specific impact these factors may have on the study results, nor does it propose specific directions and methods for follow-up studies to address these limitations.

Response 3: We rewrote the Limitations (Section 6.) and discussed the possibility that other reactions in the mouth and gut, such as glycosylation and hydrolyzation, may also affect the structure and transport of natural compounds, noting that this will require a separate in-depth review.